# Leveraging Unlabeled Data to Track Memorization

**Mahsa Forouzesh[1], Hanie Sedghi[2], Patrick Thiran[1]**
[1]École Polytechnique Fédérale de Lausanne
[2]Google Research, Brain Team
{mahsa.forouzesh, patrick.thiran}@epfl.ch, hsedghi@google.com

## Abstract

Deep neural networks may easily memorize noisy labels present in real-world data, which degrades their ability to generalize. It is therefore important to track and evaluate the robustness of models against noisy label memorization. We propose a metric, called *susceptibility*, to gauge such memorization for neural networks. Susceptibility is simple and easy to compute during training. Moreover, it does not require access to ground-truth labels and it only uses unlabeled data. We empirically show the effectiveness of our metric in tracking memorization on various architectures and datasets and provide theoretical insights into the design of the susceptibility metric. Finally, we show through extensive experiments on datasets with synthetic and real-world label noise that one can utilize susceptibility and the overall training accuracy to distinguish models that maintain a low memorization on the training set and generalize well to unseen clean data.

## 1 Introduction

Deep neural networks are prone to memorizing noisy labels in the training set, which are inevitable in many real world applications (Frénay & Verleysen, 2013; Zhang et al., 2016; Arpit et al., 2017; Song et al., 2020a; Nigam et al., 2020; Han et al., 2020; Zhang et al., 2021a; Wei et al., 2021). Given a new dataset that contains clean and noisy labels, one refers to the subset of the dataset with correct labels (respectively, with incorrect labels due to noise), as the clean (respectively, noisy) subset. When neural networks are trained on such a dataset, it is important to find the sweet spot from no fitting at all to fitting every sample. Indeed, fitting the clean subset improves the generalization performance of the model (measured by the classification accuracy on unseen clean data), but fitting the noisy subset, referred to as "memorization"[1], degrades its generalization performance. New methods have been introduced to address this issue (for example, robust architectures (Xiao et al., 2015; Li et al., 2020), robust objective functions (Li et al., 2019; Ziyin et al., 2020), regularization techniques (Zhang et al., 2017; Pereyra et al., 2017; Chen et al., 2019; Harutyunyan et al., 2020), and sample selection methods (Nguyen et al., 2019)), but their effectiveness cannot be assessed without oracle access to the ground-truth labels to distinguish the clean and the noisy subsets, or without a clean test set.

Our goal in this paper is to track memorization during training without any access to ground-truth labels. To do so, we sample a subset of the input data and label it uniformly at random from the set of all possible labels. The samples can be taken from unlabeled data, which is often easily accessible, or from the available training set with labels removed. This new held-out randomly-labeled set is created for evaluation purposes only, and does not affect the original training process.

First, we compare how different models fit the held-out randomly-labeled set after multiple steps of training on it. We observe empirically that models that have better accuracy on unseen clean test data show more resistance towards memorizing the randomly-labeled set. This resistance is captured by the number of steps required to fit the held-out randomly-labeled set. In addition, through our theoretical convergence analysis on this set, we show that models with high/low test accuracy are resistant/susceptible to memorization, respectively.

---

[1]Fitting samples that have incorrect random labels is done by memorizing the assigned label for each particular sample. Hence, we refer to it as memorization, in a similar spirit as Feldman & Zhang (2020).

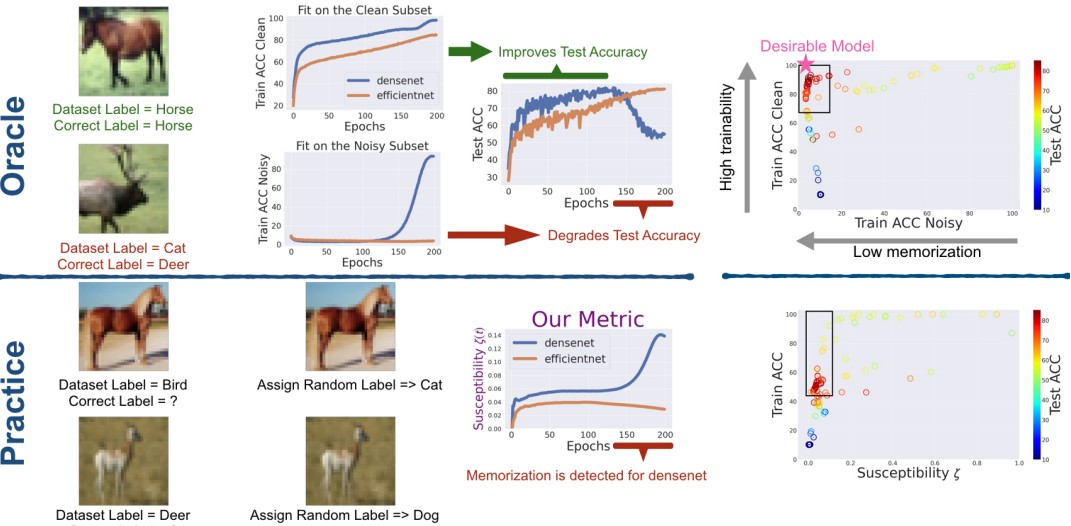

Figure 1: Models trained on CIFAR-10 with 50% label noise. **Top (Oracle access to ground-truth labels):** We observe that the fit on the clean subset of the training set (shown in the top left) and the fit on the noisy subset (located below the fit on the clean subset) affect the predictive performance (measured by the classification accuracy) on unseen clean test data differently. Fitting the clean (resp., noisy) subset improves (resp., degrades) test accuracy, as shown by the green (resp., red) arrow. With oracle access to ground-truth label, one can therefore select models with a high fit on the clean subset and a low fit on the noisy subset, as it is done in the top right to find desirable models. **Bottom (Our approach in practice):** In practice however, the ground-truth label, and hence the fit on the clean and noisy subsets are not available. In this paper, we propose the metric called susceptibility $\zeta$ to track the fit on the noisy subset of the training set. Susceptibility is computed using a mini-batch of data that are assigned with random labels independently from the dataset labels. We observe a strong correlation between susceptibility and memorization. Moreover, the susceptibility metric together with the training accuracy on the entire set, is used to recover models with low "memorization" (low fit on the noisy subset) and high "trainability" (high fit on the clean subset) without any ground-truth label oracle access. The average test accuracy of models in the top-left rectangle of the right figures are $77.93_{\pm4.68}\%$ and $76.15_{\pm6.32}\%$ for the oracle and our approach, respectively. Hence, our method successfully recovers desirable models.

Building on this result, we then propose an easy-to-compute metric that we call *susceptibility to noisy labels*, which is the difference in the objective function of a *single* mini-batch from the held-out randomly-labeled set, before and after taking an optimization step on it. At each step during training, the larger this difference is, the more the model is affected by (and is therefore susceptible to) the noisy labels in the mini-batch. Figure 1 (bottom left) provides an illustration of the susceptibility metric. We observe a strong correlation between the susceptibility and the memorization within the training set, which is measured by the fit on the noisy subset. We then show how one can utilize this metric and the overall training accuracy to distinguish models with a high test accuracy across a variety of state-of-the-art deep learning models, including DenseNet (Huang et al., 2017), EfficientNet (Tan & Le, 2019), and ResNet (He et al., 2016a) architectures, and various datasets including synthetic and real-world label noise (Clothing-1M, Animal-10N, CIFAR-10N, Tiny ImageNet, CIFAR-100, CIFAR-10, MNIST, Fashion-MNIST, and SVHN), see Figure 1 (right). Our main contributions and takeaways are summarized below:

1. We empirically observe and theoretically show that models with a high test accuracy are resistant to memorizing a randomly-labeled held-out set (Sections 2 and 5).

2. We propose the susceptibility metric, which is computed on a randomly-labeled subset of the available data. Our extensive experiments show that susceptibility closely tracks memorization of the noisy subset of the training set (Section 3).

3. We observe that models which are trainable and resistant to memorization, i.e., having a high training accuracy and a low susceptibility, have high test accuracies. We leverage this observation to propose a model-selection method in the presence of noisy labels (Section 4).

4. We show through extensive experiments that our results are persistent for various datasets, architectures, hyper-parameters, label noise levels, and label noise types (Section 6).

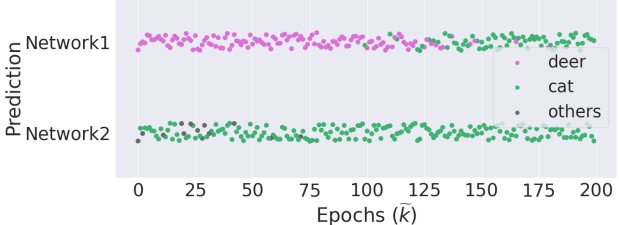

Figure 2: The evolution of output prediction of two neural networks during training on a single sample with a wrong randomly-assigned label (assigned label is "cat" while ground-truth label is "deer") versus the number of epochs (steps) $\widetilde{k}$. Network1 is a GoogLeNet (Szegedy et al., 2015) (pre-)trained on CIFAR-10 dataset with clean labels and has initial test accuracy of $95.36\%$. Network2 is a GoogLeNet (pre-)trained on CIFAR-10 dataset with $50\%$ label noise level and has initial test accuracy of $58.35\%$. Network2 has a lower test accuracy compared to Network1, and we observe that it memorizes this new sample after only $\widetilde{k} = 2$ steps. In contrast, Network1 persists in predicting the correct output for this sample for longer, and memorizes the new sample only after $\widetilde{k} = 99$ steps. More examples and an ablation study on the effect of calibration are given in Figures 12 and 14 in Appendix D, respectively.

**Related work** Garg et al. (2021a) show that for models trained on a mixture of clean and noisy data, a low accuracy on the noisy subset of the training set and a high accuracy on the clean subset of the training set guarantee a low generalization error. With oracle access to the ground-truth labels to compute these accuracies, one can therefore predict which models perform better on unseen clean data. This is done in Figure 1 (Oracle part). However, in practice, there is no access to ground-truth labels. Our work therefore complements the results of (Garg et al., 2021a) by providing a practical approach (see Figure 1 (Practice part)). Moreover, both our work and (Zhang et al., 2019c) emphasize that a desirable model differentiates between fitting clean and noisy samples. Zhang et al. (2019c) showcase this intuition by studying the drop in the training accuracy of models trained when label noise is injected in the dataset. We do it by studying the resistance/susceptibility of models to noisy labels of a held-out set. Zhang et al. (2019c) only study the training accuracy drop for settings where the available training set is clean. However, when the training set is itself noisy, which is the setting of interest in our work, we observe that this training accuracy drop does not predict memorization, unlike our susceptibility metric (see Figure 9 (left) in Appendix A). Furthermore, even though the metric proposed in Lu & He (2022) is rather effective as an early stopping criterion, it is not able to track memorization, contrary to susceptibility (see Figure 9 (middle)). For a thorough comparison to other related work refer to Appendix A.

## 2 GOOD MODELS ARE RESISTANT TO MEMORIZATION

Consider models $f_{\mathbf{W}}(\mathbf{x})$ with parameter matrix $\mathbf{W}$ trained on a dataset $S = \{(\mathbf{x}_i, y_i)\}_1^n$, which is a collection of $n$ input-output pairs that are drawn from a data distribution $\mathcal{D}$ over $\mathcal{X} \times \mathcal{Y}$ in a multi-class classification setting. We raise the following question: How much are these models resistant/susceptible to memorization of a new low-label-quality (noisy) dataset when they are trained on it? Intuitively, we expect a model with a high accuracy on correct labels to stay persistent on its predictions and hence to be *resistant* to the memorization of this new noisy dataset. We use the number of steps that it requires to fit the noisy dataset, as a measure for this resistance. The larger this number, the stronger the resistance (hence, the lower the susceptibility). In summary, we conjecture that models with high test accuracy on unseen clean data are more resistant to memorization of noisy labels, and hence take longer to fit a randomly-labeled held-out set.

To mimic a noisy dataset, we create a set with random labels. More precisely, we define a randomly-labeled held-outset of samples $\widetilde{S} = \{(\widetilde{\mathbf{x}}_i, \widetilde{y}_i)\}_1^{\widetilde{n}}$, which is generated from $\widetilde{n}$ unlabeled samples $\{\widetilde{\mathbf{x}}_i\}_1^{\widetilde{n}} \sim \mathcal{X}$ that are either accessible, or are a subset of $\{\mathbf{x}_i\}_1^n$. The outputs $\widetilde{y}_i$ of dataset $\widetilde{S}$ are drawn independently (both from each other and from labels in $S$) and uniformly at random from the set of possible classes. Therefore, to fit $\widetilde{S}$, the model needs to memorize the labels. We track the fit (memorization) of a model on $\widetilde{S}$ after $\widetilde{k}$ optimization steps on the noisy dataset $\widetilde{S}$.

We perform the following empirical test. In Figure 2, we compare two deep neural networks. The first one has a high classification accuracy on unseen clean test data, whereas the other one has a low

test accuracy. We then train both these models on a single sample with an incorrect label ($\widetilde{S}$ contains a single incorrectly-labeled sample). We observe that the model with a high test accuracy takes a long time (in terms of number of training steps) to fit this sample; hence this model is *resistant* to memorization as the number of steps $\widetilde{k}$ required to fit $\widetilde{S}$ is large. In contrast, the model with a low test accuracy memorizes this sample after taking only a few steps; hence this model is *susceptible* to memorization as $\widetilde{k}$ required to fit $\widetilde{S}$ is small[2]. This observation reinforces our intuition, that *good models*, measured by a high test accuracy, *are more resistant to memorization of noisy labels*.

Next, we further validate this observation theoretically by performing a convergence analysis on the held-out set $\widetilde{S}$, where the input samples $\{\widetilde{\mathbf{x}}_i\}_1^{\widetilde{n}}$ of $\widetilde{S}$ are the same as the inputs of dataset $S$ (and hence $\widetilde{n} = n$). We consider the binary-classification setting. The output vector $\widetilde{\mathbf{y}} = (\widetilde{y}_1, \cdots, \widetilde{y}_n)^T$ is a vector of independent random labels that take values uniformly in $\{-1, 1\}$. Therefore, creating this dataset $\widetilde{S}$ does not require extra information, and in particular no access to the data distribution $\mathcal{D}$. Consider the network $f_{\mathbf{W}}(\mathbf{x})$ as a two-layer neural network with ReLU non-linearity (denoted by $\sigma(x) = \max\{x, 0\}$) and $m$ hidden-units:

$$f_{\mathbf{W}}(\mathbf{x}) = \frac{1}{\sqrt{m}} \sum_{r=1}^{m} a_r \sigma\left(\mathbf{w}_r^T \mathbf{x}\right),$$

where $\mathbf{x} \in \mathbb{R}^d$ is the input vector, $\mathbf{W} = (\mathbf{w}_1, \cdots, \mathbf{w}_m) \in \mathbb{R}^{d \times m}$, $\mathbf{a} = (a_1, \cdots, a_m) \in \mathbb{R}^m$ are the weight matrix of the first layer, and the weight vector of the second layer, respectively. For simplicity, the outputs associated with the inputs $\{\mathbf{x}_i\}_1^n$ are denoted by vector $\mathbf{f}_{\mathbf{W}} \in \mathbb{R}^n$ instead of $\{f_{\mathbf{W}}(\mathbf{x}_i)\}_1^n$. The first layer weights are initialized as $\mathbf{w}_r(0) \sim \mathcal{N}\left(\mathbf{0}, \kappa^2 \mathbf{I}\right) \forall r \in [m]$, where $0 < \kappa \leq 1$ is the magnitude of initialization and $\mathcal{N}$ denotes the normal distribution. $a_r$s are independent random variables taking values uniformly in $\{-1, 1\}$, and are considered to be fixed throughout training.

We define the objective function on datasets $S$ and $\widetilde{S}$, respectively, as

$$\Phi(\mathbf{W}) = \frac{1}{2} \|\mathbf{f}_{\mathbf{W}} - \mathbf{y}\|_2^2 = \frac{1}{2} \sum_{i=1}^{n} (f_{\mathbf{W}}(\mathbf{x}_i) - y_i)^2, \qquad \widetilde{\Phi}(\mathbf{W}) = \frac{1}{2} \sum_{i=1}^{n} (f_{\mathbf{W}}(\mathbf{x}_i) - \widetilde{y}_i)^2. \qquad (1)$$

Label noise level (LNL) of $S$ (and accordingly of the label vector $\mathbf{y}$) is the ratio $n_1/n$, where $n_1$ is the number of samples in $S$ that have labels that are independently drawn uniformly in $\{-1, 1\}$, and the remaining $n - n_1$ samples in $S$ have the ground-truth label. The two extremes are $S$ with LNL = 0, which is the clean dataset, and $S$ with LNL = 1, which is a dataset with entirely random labels.

To study the convergence of different models on $\widetilde{S}$, we compute the objective function $\widetilde{\Phi}(\mathbf{W})$ after $\tilde{k}$ steps of training on the dataset $\widetilde{S}$. Therefore, the overall training/evaluation procedure is as follows; for $r \in [m]$, the second layer weights of the neural network are updated according to gradient descent:

$$\mathbf{w}_r(t+1) - \mathbf{w}_r(t) = -\eta \frac{\partial \overline{\Phi}(\mathbf{W}(t))}{\partial \mathbf{w}_r},$$

where for $0 \leq t < k$: $\overline{\Phi} = \Phi$, and for $k \leq t < k + \tilde{k}$: $\overline{\Phi} = \widetilde{\Phi}$, whereas $\eta$ is the learning rate. Note that we refer to $\Phi(\mathbf{W}(t))$ as $\Phi(t)$ and to $\widetilde{\Phi}(\mathbf{W}(t))$ as $\widetilde{\Phi}(t)$.

When a model fits the set $\widetilde{S}$, the value of $\widetilde{\Phi}$ becomes small, say below some threshold $\varepsilon$. The resistance of a model to memorization (fit) on the set $\widetilde{S}$ is then measured by the number of steps $\widetilde{k}^*(\varepsilon)$ such that $\widetilde{\Phi}(k + \widetilde{k}) \leq \varepsilon$ for $\widetilde{k} > \widetilde{k}^*(\varepsilon)$. The larger (respectively, smaller) $\widetilde{k}^*$ is, the more the model is *resistant* (resp, susceptible) to this memorization. We can reason on the link between good model and memorization from the following proposition.

**Proposition 1** (Informal)**.** *The objective function $\widetilde{\Phi}$ at step $k + \widetilde{k}$ is a decreasing function of the label noise level (LNL) in $S$, and of the number of steps $\widetilde{k}$.*

---

[2]Interestingly, we observe that the situation is different for a correctly-labeled sample. Figure 13 in Appendix D shows that models with a higher test accuracy typically fit an unseen correctly-labeled sample faster than models with a lower test accuracy.

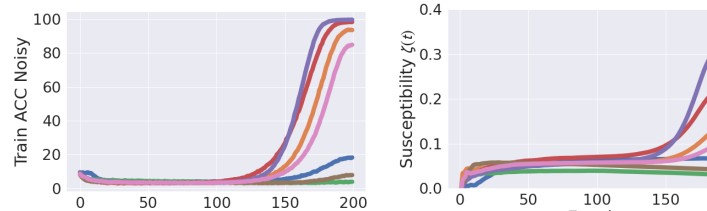

Figure 3: Accuracy on the noisy subset of the training set versus susceptibility $\zeta(t)$ (Equation (2)) for deep convolutional neural networks (ranging from a $5-$layer cnn to more sophisticated structures such as EfficientNet (Tan & Le, 2019)) trained on CIFAR-10 with 50% label noise. We observe a strong Pearson correlation coefficient between Train ACC Noisy and $\zeta(t)$: $\rho = 0.884$.

Proposition 1 yields that models trained on $S$ with a low LNL (which are models with a high test accuracy on clean data) push $\widetilde{\Phi}(k + \widetilde{k})$ to become large. The number of steps $\widetilde{k}^*(\varepsilon)$ should therefore be large for $\widetilde{\Phi}$ to become less than $\epsilon$: these models resist to the memorization of $\widetilde{S}$. Conversely, models trained on $S$ with a high LNL (which are models with a low test accuracy), allow for $\widetilde{k}^*(\varepsilon)$ to be small and give up rapidly to the memorization of $\widetilde{S}$. These conclusions match the empirical observation in Figure 2, and therefore further support the intuition that *good models are resistant to memorization*. Please refer to Section 5 for the formal theoretical results that produce Proposition 1.

## 3 EVALUATING RESISTANCE TO MEMORIZATION

In Section 2 we showed that desirable models in terms of high classification accuracy on unseen clean data are resistant to the memorization of a randomly-labeled set. In this section, we describe a simple and computationally efficient metric to measure this resistance. To evaluate it efficiently at each step of the forward pass, we propose to take a *single* step on *multiple* randomly labeled samples, instead of taking *multiple* steps on a *single* randomly labeled sample (as done in Figure 2). To make the metric even more computationally efficient, instead of the entire randomly labeled set $\widetilde{S}$ (as in our theoretical developments), we only take a single mini-batch of $\widetilde{S}$. For simplicity and with some abuse of notation, we still refer to this single mini-batch as $\widetilde{S}$.

To track the prediction of the model over multiple samples, we compare the objective function on $\widetilde{S}$ before and after taking a single optimization step on it. The learning rate, and its schedule, have a direct effect on this single optimization step. For certain learning rate schedules, the learning rate might become close to zero (e.g., at the end of training), and hence the magnitude of this single step would become too small to be informative. To avoid this, and to account for the entire history of the optimization trajectory[3], we compute the average difference over the learning trajectory (a moving average). We propose the following metric, which we call *susceptibility to noisy labels*, $\zeta(t)$. At step $t$ of training,

$$\zeta(t) = \frac{1}{t} \sum_{\tau=1}^{t} \left[ \widetilde{\Phi}\left(\mathbf{W}(\tau)\right) - \widetilde{\Phi}\left(\widetilde{\mathbf{W}}(\tau+1)\right) \right],$$
(2)

where $\mathbf{W}(\tau)$, $\widetilde{\mathbf{W}}(\tau+1)$ are the model parameters before and after a single update step on $\widetilde{S}$. Algorithm 1 describes the computation of $\zeta$.

---

**Algorithm 1** Computes the susceptibility to noisy labels $\zeta$

---

1: **Input:** Dataset $S$, Number of Epochs $T$
2: Sample a mini-batch $\widetilde{S}$ from $S$; Replace its labels with random labels
3: Initialize network $f_{\mathbf{W}(0)}$
4: Initialize $\zeta(0) = 0$
5: **for** $t = 1, \cdots, T$ **do**
6:     Update $f_{\mathbf{W}(t)}$ from $f_{\mathbf{W}(t-1)}$ using dataset $S$
7:     Compute $\widetilde{\Phi}(\mathbf{W}(t))$
8:     Update $f_{\widetilde{\mathbf{W}}(t+1)}$ from $f_{\mathbf{W}(t)}$ using dataset $\widetilde{S}$
9:     Compute $\widetilde{\Phi}(\widetilde{\mathbf{W}}(t+1))$
10:     Compute $\zeta(t) = \frac{1}{t}\Big[(t-1)\zeta(t-1) + \widetilde{\Phi}(\mathbf{W}(t)) - \widetilde{\Phi}(\widetilde{\mathbf{W}}(t+1))\Big]$
11: **end for**
12: **Return** $\zeta$.

---

[3]The importance of the entire learning trajectory including its early phase is emphasized in prior work such as (Jastrzebski et al., 2020).

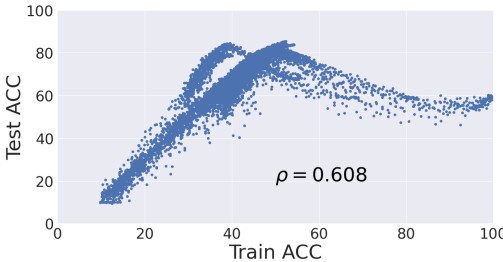 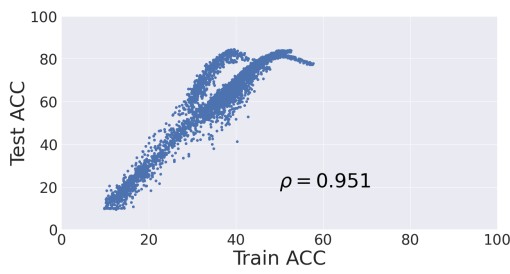

(a) Test accuracy versus train accuracy for all models at all epochs

(b) Test accuracy versus train accuracy for models that have $\zeta(t) \leq 0.05$

Figure 4: The effect of filtering the models based on the susceptibility metric $\zeta(t)$ for models trained on CIFAR-10 with $50\%$ label noise. We observe that by removing models with a high value of $\zeta(t)$, the correlation between the training accuracy (accessible to us in practice) and the test accuracy (not accessible) increases a lot, and we observe that among the selected models, a higher training accuracy implies a higher test accuracy. Refer to Figures 16, 18, 21, 24, 26 for similar results on 5 other datasets. We use a threshold such that half of the models have $\zeta <$ threshold and hence remain after filtration.

The lower $\zeta(t)$ is, the less the model changes its predictions for $\widetilde{S}$ after a single step of training on it and thus the less it memorizes the randomly-labeled samples. We say therefore that a model is resistant (to the memorization of randomly-labeled samples) when the susceptibility to noisy labels $\zeta(t)$ is low.

**Susceptibility $\zeta$ tracks memorization for different models**    The classification accuracy of a model on a set is defined as the ratio of the number of samples where the label predicted by the model matches the label on the dataset, to the total number of samples. We refer to the subset for which the dataset label is different from the ground-truth label as the noisy subset of the training set. Its identification requires access to the ground-truth label. Recall that we refer to memorization within the training set as the fit on the noisy subset, which is measured by the accuracy on it. We call this accuracy "Train ACC Noisy" in short. In Figure 3, we observe a strong positive correlation between the memorization of the held-out randomly labeled set, which is tracked by $\zeta(t)$ (Equation (2)) and the memorization of noisy labels within the training set, tracked by Train ACC Noisy. Susceptibility $\zeta$, which is computed on a mini-batch with labels independent from the training set, can therefore predict the resistance to memorization of the noisy subset of the training set. In Figure 36, we show the robustness of susceptibility $\zeta$ to the mini-batch size, and to the choice of the particular mini-batch.

**Susceptibility $\zeta$ tracks memorization for a single model**    It is customary when investigating different capabilities of a model, to keep checkpoints/snapshots during training and decide which one to use based on the desired property, see for example (Zhang et al., 2019a; Chatterji et al., 2019; Neyshabur et al., 2020; Andreassen et al., 2021; Baldock et al., 2021). With access to ground-truth labels, one can track the fit on the noisy subset of the training set to find the model checkpoint with the least memorization, which is not necessarily the end checkpoint, and hence an early-stopped version of the model. However, the ground-truth label is not accessible in practice, and therefore the signals presented in Figure 3 (left) are absent. Moreover, as discussed in Figure 7, the training accuracy on the entire training set is also not able to recover these signals. Using susceptibility $\zeta$, we observe that these signals can be recovered without any ground-truth label access, as shown in Figure 3 (right). Therefore, $\zeta$ can be used to find the model checkpoint with the least memorization. For example, in Figure 3, for the ResNet model, susceptibility $\zeta$ suggests to select model checkpoints before it sharply increases, which exactly matches with the sharp increase in the fit on the noisy subset. On the other hand, for the MobileNet model, susceptibility suggests to select the end checkpoint, which is also consistent with the selection according to the fit on the noisy subset.

## 4  GOOD MODELS ARE RESISTANT AND TRAINABLE

When networks are trained on a dataset that contains clean and noisy labels, the fit on the clean and noisy subsets affect the generalization performance of the model differently. Therefore, as we observe in Figure 4a for models trained on a noisy dataset (details are deferred to Appendix B), the

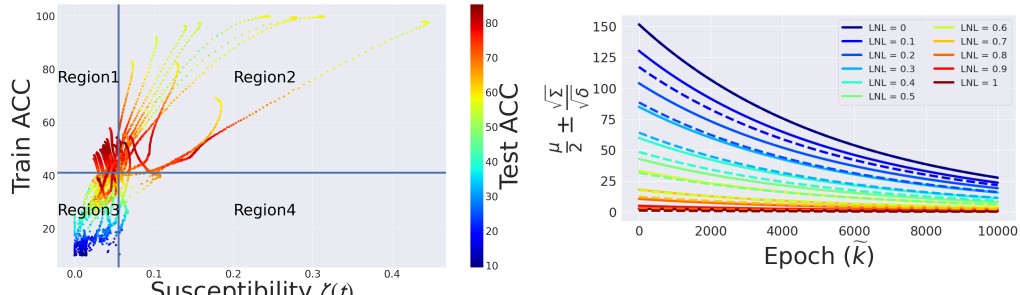

Figure 5: For models trained on CIFAR-10 with 50% label noise, using the average values of $\zeta(t)$ and of the training accuracy over the available models we can obtain 4 different regions. **Region 1**: Trainable and resistant, with average test accuracy of **76**%. **Region 2**: Trainable but not resistant, with average test accuracy of 70.59%. **Region 3**: Not trainable but resistant, with average test accuracy of 52.49%. **Region 4**: Neither trainable nor resistant, with average test accuracy of 58.23%. We observe that the models in Region 1 generalize well on unseen data, as they have a high test accuracy.

Figure 6: The lower (dashed lines) and upper (solid lines) bound terms of Theorem 2 that depend on the label noise level (LNL). They are computed from the eigenvectors and eigenvalues of the Gram-matrix for 1000 samples of the MNIST dataset, computed over 10 random draws computed with hyper-parameters $\eta = 10^{-6}$, $k = 10000$ and $\delta = 0.05$. We observe the bounds in Theorem 2 are a decreasing function of LNL.

correlation between the training accuracy (accuracy on the entire training set) and the test accuracy (accuracy on the unseen clean test set) is low. Interestingly however, we observe in Figure 4b that if we remove models with a high memorization on the noisy subset, as indicated by a large susceptibility to noisy labels $\zeta(t)$, then the correlation between the training and the test accuracies increases significantly (from $\rho = 0.608$ to $\rho = 0.951$), and the remaining models with low values of $\zeta(t)$ have a good generalization performance: a higher training accuracy implies a higher test accuracy. This is especially important in the practical settings where we do not know how noisy the training set is, and yet must reach a high accuracy on clean test data.

After removing models with large values of $\zeta(t)$, we select models with a low label noise memorization. However, consider an extreme case: a model that always outputs a constant is also low at label noise memorization, but does not learn the useful information present in the clean subset of the training set either. Clearly, the models should be "trainable" on top of being resistant to memorization. A "trainable model" is a model that has left the very initial stages of training, and has enough capacity to learn the information present in the clean subset of the dataset to reach a high accuracy on it. The training accuracy on the entire set is a weighted average of the training accuracy on the clean and noisy subsets. Therefore, among the models with a low accuracy on the noisy subset (selected using susceptibility $\zeta$ in Figure 4b), we should further restrict the selection to those with a high overall training accuracy, which corresponds to a high accuracy on the clean subset.

We can use the two metrics, susceptibility to noisy labels $\zeta(t)$ and overall training accuracy, to partition models that are being trained on some noisy dataset in four different regions using their average values, as shown in Figure 5: Region 1: Trainable and resistant to memorization, Region 2: Trainable but not resistant, Region 3: Not trainable but resistant, and Region 4: Neither trainable nor resistant. Note that the colors of each point, which indicate the value of the test accuracy are only for illustration and are not used to find these regions. The average test accuracy is the highest for models in Region 1 (i.e., that are resistant to memorization on top of being trainable). In particular, going from Region 2 to Region 1 increases the average test accuracy of the selected models from 70.59% to 76%, i.e., a 7.66% relative improvement in the test accuracy. This improvement is consistent with other datasets as well: 6.7% for Clothing-1M (Figure 29a), 4.2% for Animal-10N (Figure 29b), 2.06% for CIFAR-10N (Figure 29c), 31.4% for noisy MNIST (Figure 15), 33% for noisy Fashion-MNIST (Figure 17), 33.6% for noisy SVHN (Figure 19), 15.1% for noisy CIFAR-100 (Figure 22), and 8% for noisy Tiny ImageNet (Figure 25) datasets (refer to Appendix E for detailed results).

**Comparison with a noisy validation set:** Given a dataset containing noisy labels, a correctly-labeled validation set is not accessible without ground-truth label access. One can however use a subset of the available dataset and assess models using this noisy validation set. An important advantage of our approach compared to this assessment is the information that it provides about the memorization within the training set, which is absent from performance on a test or a noisy

validation set. For example, in Figures 34a and 35a we observe models (on the very top right) that have memorized $100\%$ of the noisy subset of the training set, and yet have a high test accuracy. However, some models (on the top middle/left part), which have the same fit on the clean subset but lower memorization, have lower or similar test accuracy. Even the test accuracy was therefore not able to compare these models in terms of memorization. For more details, refer to Appendix C.

Moreover, as formally discussed in (Lam & Stork, 2003), the number of noisy validation samples that are equivalent to a single clean validation sample depends on the label noise level of the dataset and on the true error rate of the model, which are both unknown in practice, making it difficult to pick an acceptable size for the validation set without compromising the training set. If we use a small validation set (which allows for a large training set size), then we observe a very low correlation between the validation and the test accuracies (as reported in Table 1 and Figure 10 in Appendix C). In contrast, our approach provides much higher correlation to the test accuracy. We also conducted experiments with noisy validation sets with larger sizes and observed that the size of the noisy validation set would need to be increased around ten-fold to reach the same correlation as our approach. Increasing the size of the validation set removes data out of the training set, which may degrade the overall performance, whereas our approach leaves the entire available dataset for training.

## 5 CONVERGENCE ANALYSIS

In this section, we elaborate on the analysis that leads to (the informal) Proposition 1. A matrix, a vector and a scalar are denoted respectively by $\mathbf{A}, \mathbf{a}$, and $a$. The identity matrix is denoted by $\mathbf{I}$. The indicator function of a random event $A$ is denoted by $\mathbb{I}(A)$. The $(i, j)-$th entry of Matrix $\mathbf{A}$ is $\mathbf{A}_{ij}$.

Consider the setting described in Section 2 with $\|\mathbf{x}\|_2 = 1$ and $|y| \leq 1$ for $(\mathbf{x}, y) \sim \mathcal{D} \coloneqq \mathcal{X} \times \mathcal{Y}$, where $\mathcal{X} = \mathbb{R}^d$ and $\mathcal{Y} = \mathbb{R}$. For input samples $\{\mathbf{x}_i\}_1^n$, the $n \times n$ Gram matrix $\mathbf{H}^\infty$ has entries

$$\mathbf{H}_{ij}^\infty = \mathbb{E}_{\mathbf{w} \sim \mathcal{N}(\mathbf{0}, \mathbf{I})} \left[ \mathbf{x}_i^T \mathbf{x}_j \mathbb{I}\{\mathbf{w}^T \mathbf{x}_i \geq 0, \mathbf{w}^T \mathbf{x}_j \geq 0\} \right] = \frac{\mathbf{x}_i^T \mathbf{x}_j \left( \pi - \arccos(\mathbf{x}_i^T \mathbf{x}_j) \right)}{2\pi}, \ \forall i, j \in [n], \quad (3)$$

and its eigen-decomposition is $\mathbf{H}^\infty = \sum_{i=1}^n \lambda_i \mathbf{v}_i \mathbf{v}_i^T$, where the eigenvectors $\mathbf{v}_i \in \mathbb{R}^n$ are orthonormal and $\lambda_0 = \min\{\lambda_i\}_1^n$. Using the eigenvectors and eigenvalues of $\mathbf{H}^\infty$, we have the following theorem.

**Theorem 1.** *For $\kappa = O\left( \frac{\epsilon \sqrt{\delta} \lambda_0}{n^{3/2}} \right)$, $m = \Omega\left( \frac{n^9}{\lambda_0^6 \epsilon^2 \kappa^2 \delta^4} \right)$, and $\eta = O\left( \frac{\lambda_0}{n^2} \right)$, with probability at least $1 - \delta$ over the random parameter initialization described in Section 2, we have:*

$$\left\| \mathbf{f}_{\mathbf{W}(k+\tilde{k})} - \widetilde{\mathbf{y}} \right\|_2 = \sqrt{\sum_{i=1}^n \left[ \mathbf{v}_i^T \mathbf{y} - \mathbf{v}_i^T \widetilde{\mathbf{y}} - (1 - \eta \lambda_i)^k \mathbf{v}_i^T \mathbf{y} \right]^2 (1 - \eta \lambda_i)^{2\tilde{k}}} \pm \epsilon.$$

The proof is provided in Appendix J.

Theorem 1 enables us to approximate the objective function on dataset $\widetilde{S}$ (Equation (1)) as follows:

$$\widetilde{\Phi}(k + \tilde{k}) \approx \frac{1}{2} \sum_{i=1}^n \left[ p_i - \widetilde{p}_i - (1 - \eta \lambda_i)^k p_i \right]^2 (1 - \eta \lambda_i)^{2\tilde{k}}, \quad (4)$$

where $p_i = \mathbf{v}_i^T \mathbf{y}$ and $\widetilde{p}_i = \mathbf{v}_i^T \widetilde{\mathbf{y}}$. Let us define

$$\mu = \sum_{i=1}^n \mathbb{E}\left[ p_i^2 \right] \left[ 1 - (1 - \eta \lambda_i)^k \right]^2 (1 - \eta \lambda_i)^{2\tilde{k}}, \quad \Sigma = \text{Var}_{\widetilde{\mathbf{p}}, \mathbf{p}} \left[ \widetilde{\Phi}(k + \tilde{k}) \right]. \quad (5)$$

We numerically observe that $\mu/2 \pm \sqrt{\Sigma/\delta}$ are both decreasing functions of the label noise level (LNL) of the label vector $\mathbf{y}$, and of $\widetilde{k}$ (see Figures 6, and 40, 41, 42, 43 in Appendix K for different values of the learning rate $\eta$, number of steps $k$, and datasets). The approximation in Equation (4) together with the above observation then yields the following lower and upper bounds for $\widetilde{\Phi}(k + \tilde{k})$.

**Theorem 2.** *With probability at least $1 - \delta$ over the draw of the random label vector $\widetilde{\mathbf{y}}$, given the approximation made in Equation (4),*

$$\frac{\mu}{2} - \sqrt{\frac{\Sigma}{\delta}} \leq \widetilde{\Phi}(k + \tilde{k}) - \frac{1}{2} \sum_{i=1}^n (1 - \eta \lambda_i)^{2\tilde{k}} \leq \frac{\mu}{2} + \sqrt{\frac{\Sigma}{\delta}} \quad (6)$$

The proof is provided in Appendix K. Because the term $\sum_{i=1}^{n}(1-\eta\lambda_i)^{2\tilde{k}}$ is independent of LNL of the label vector $\mathbf{y}$, we can conclude that $\widetilde{\Phi}(k+\tilde{k})$ is a decreasing function of LNL. Moreover, Since $0 < 1 - \eta\lambda_i < 1$, the term $\sum_{i=1}^{n}(1-\eta\lambda_i)^{2\tilde{k}}$ is also a decreasing function of $\tilde{k}$. Therefore, similarly, we can conclude that $\widetilde{\Phi}(k+\tilde{k})$ is a decreasing function of $\tilde{k}$. Proposition 1 summarizes this result.

## 6    ON THE GENERALITY OF THE OBSERVED PHENOMENA

In this section, we provide additional experiments that show our empirical results hold across different choices of dataset, training and architecture design as well as label noise level and form.

**Dataset:** In Appendix E.1, we provide a walk-through on how to use our method for real-world noisy-labeled datasets Clothing-1M, Animal-10N, and CIFAR-10N. Our results consistently hold for MNIST, Fashion MNIST, SVHN, CIFAR-100, and Tiny ImageNet datasets; see Figures 15-29.

**Learning rate schedule:** Our results are not limited to a specific optimization scheme. In our experiments, we apply different learning rate schedules, momentum values, and optimizers (SGD and Adam) (for details see Appendix B). More specifically, we show in Figure 31 (in Appendix F) that the strong correlation between memorization and our metric $\zeta(t)$ stays consistent for both learning rate schedulers `cosineannealing` and `exponential`.

**Architecture:** Results of Sections 3 and 4 are obtained from a variety of architecture families, such as DenseNet (Huang et al., 2017), MobileNet (Howard et al., 2017), VGG (Simonyan & Zisserman, 2014), and ResNet (He et al., 2016a). For the complete list of architectures, see Appendix B. We observe that $\zeta(t)$ does not only detect resistant architecture families (as done for example in Figure 3), but that it is also able to find the best design choice (e.g., width) among configurations that are already resistant, see Figure 30 in Appendix F.

**Low label noise levels:** For models trained on CIFAR-10 and CIFAR-100 datasets with $10\%$ label noise (instead of $50\%$), we still observe a high correlation between accuracy on the noisy subset and $\zeta(t)$ in Figures 32 and 33 in Appendix F. Moreover, we observe in Figures 34 and 35 in Appendix F that the average test accuracy of the selected models using our metric is comparable with the average test accuracy of the selected models with access to the ground-truth label.

**Asymmetric label noise:** In addition to the real-world label noises and synthetic symmetric label noise, we have also performed experiments with synthetic asymmetric label noise as proposed in (Xia et al., 2021). Using our approach, the average test accuracy of the selected models is $66.307\%$, whereas the result from oracle is $66.793\%$ (see Figure 38 in Appendix E).

## 7    CONCLUSION

We have proposed a simple but surprisingly effective approach to track memorization of the noisy subset of the training set using a single mini-batch of unlabeled data. Our contributions are three-fold. First, we have shown that models with a high test accuracy are resistant to memorization of a held-out randomly-labeled set. Second, we have proposed a metric, susceptibility, to efficiently measure this resistance. Third, we have empirically shown that one can utilize susceptibility and the overall training accuracy to distinguish models (whether they are a single model at various checkpoints, or different models) that maintain a low memorization on the training set and generalize well to unseen clean data while bypassing the need to access the ground-truth label. We have studied model selection in a variety of experimental settings and datasets with label noise, ranging from selecting the "best" models from "good" models (for easy datasets such as Animal-10N) to selecting "good" models from "bad" models (for more complex datasets such as Clothing-1M).

Our theoretical results have direct implications for online settings. When the quality of the labels for new data stream is low, Theorem 1 can directly compare how different models converge on this low quality data, and help in selecting resistant models, which converge slowly on this data. Our empirical results provide new insights on the way models memorize the noisy subset of the training set: the process is similar to the memorization of a purely randomly-labeled held-out set. Finally, an important direction for future work is to study how susceptibility performs as a regularizer.

**Ethics Statement:**   This work is a foundational investigation into the theory of deep learning. We believe our submission does not raise ethical concerns. We use open public datasets and implementations. Our study does not involve human subjects, potentially harmful insights, potential conflicts of interest and sponsorship, discrimination/bias/fairness concerns, privacy and security issues, legal compliance, and research integrity issues.

**Reproducibility Statement:**   The source code to reproduce our results is available in the supplementary material. All the datasets and architectures used in this paper are publicly available and are properly cited. Reproducibility of all our experiments is ensured by providing the experimental setup details in Appendix B. The data processing steps are provided both in Appendix B and in the source code. The assumptions and preliminaries needed for the proofs of our theoretical results are provided in Appendix G. The proofs are provided in Appendices H, I, J, and K.

**Acknowledgement:**   We thank Chiyuan Zhang and Behnam Neyshabur for their valuable feedback on the draft of this work. We would also like to thank anonymous reviewers, area chairs, and program chairs for their constructive comments, which helped us to improve the quality of this paper.

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

# A  ADDITIONAL RELATED WORK

**Memorization**    Arpit et al. (2017) describe memorization as the behavior shown by neural networks when trained on noisy data. Patel & Sastry (2021) define memorization as the difference in the predictive performance of models trained on the clean data and on the noisy data distributions, and propose a robust objective function that resists to memorization. The effect of neural network architecture on their robustness to noisy labels is studied in (Li et al., 2021), which measure robustness to noisy labels by the prediction performance of the learned representations on the ground-truth target function. Feldman & Zhang (2020) formally define memorization of a sample for an algorithm as the inability of the model to predict the output of a sample based on the rest of the dataset, so that the only way that the model can fit the sample is by memorizing its label. Our definition of memorization as the fit on the noisy subset of the training set follows the same principle; the fit of a randomly-labeled individual sample is only possible if the totally random label associated with the sample is memorized, and it is impossible to predict using the rest of the dataset.

**Learning before Memorization**    Multiple studies have reported that neural networks learn simple patterns first, and memorize the noisy labeled data later in the training process (Arpit et al., 2017; Gu & Tresp, 2019; Krueger et al., 2017; Liu et al., 2020). Hence, early stopping might be useful when learning with noisy labels. Parallel to our study, Lu & He (2022) propose early stopping metrics that are computed on the training set. These early stopping metrics are computed by an iterative algorithm (either a GMM or k-Means) on top of the training loss histograms. This adds computational overhead compared to our approach, which is based on a simple metric that can be computed on the fly. Moreover, we can observe in Figure 9 that the metric proposed in Lu & He (2022), which is the mean difference between distributions obtained from the GMM applied on top of the training losses, does not correlated with memorization on the training set, as opposed to our metric susceptibility. Hence, we can conclude that, even though the mean difference metric might be a rather well early stopping criterion for a single setting, it does not perform well in terms of comparing different settings to one another. Moreover, Rahaman et al. (2019); Xu et al. (2019a;b) show that neural networks learn lower frequencies in the input space first and higher frequencies later. However, the monotonic behaviour of neural networks during the training procedure is recently challenged by the epoch-wise double descent (Zhang et al., 2020; Nakkiran et al., 2021). When a certain amount of fit is observed on the entire training set, how much of this fit corresponds to the fit on the clean and noisy subsets, respectively, is still unclear. In this paper, we propose an approach to track the fit to the clean subset and the fit to the noisy subset (memorization) explicitly.

**Training Speed**    Lyle et al. (2020) show there is a connection between training speed and the marginal likelihood for linear models. Jiang & Gal also find an explanation for the connection between training speed and generalization, and Ru et al. (2021) use this connection for neural architecture search. However, in Figure 7, we observe that a sharp increase in the training accuracy (a high training speed), does not always indicate an increase in the value of accuracy on the noisy subset. This result suggests that studies that relate training speed with generalization (Lyle et al., 2020; Ru et al., 2021) might not be extended as such to noisy-label training settings. On the other hand, we observe a strong correlation between susceptibility $\zeta$ and accuracy on the noisy subset.

**Leveraging Unlabeled/Randomly-labeled Data**    Unlabeled data has been leveraged previously to predict out-of-distribution performance (when there is a mismatch between the training and test distributions) (Garg et al., 2021b). Li et al. (2019) introduce synthetic noise to unlabeled data, to propose a noise-tolerant meta-learning training approach. Zhang et al. (2021b) use randomly labeled data to perform neural architecture search. In our paper, we leverage unlabeled data to track memorization of label noise.

**Benefits of Memorization**    Studies on long-tail data distributions show potential benefits from memorization when rare and atypical instances are abundant in the distribution (Feldman, 2020; Feldman & Zhang, 2020; Brown et al., 2021). These studies argue that the memorization of these rare instances is required to guarantee low generalization error. On a separate line of work, previous empirical observations suggest that networks trained on noisy labels can still induce good representations from the data, despite their poor generalization performance (Li et al., 2020; Maennel et al., 2020). In this work, we propose an approach to track memorization when noisy labels are present.

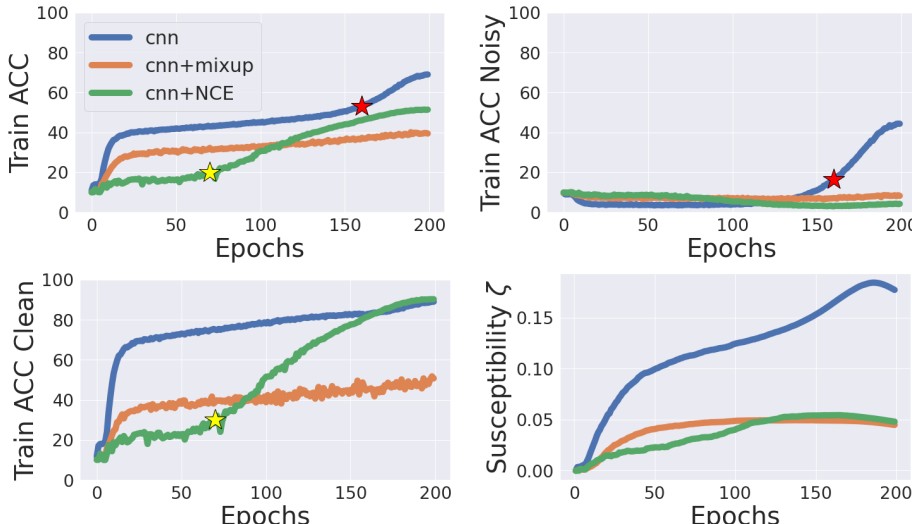

Figure 7: Accuracy on the entire, the clean and the noisy subsets of the training set, versus susceptibility in Equation (2) for a $5-$layer convolutional neural network trained on CIFAR-10 with $50\%$ label noise without any regularization, with Mixup (Zhang et al., 2017), and with Active Passive loss using the normalized cross entropy together with reverse cross entropy loss (denoted by NCE) (Ma et al., 2020). The sharp increases in the overall training accuracy (illustrated by the red and yellow stars) can be caused by an increase in the accuracy of either the clean (the yellow star) and noisy (the red star) subsets. Therefore, this sharp increase could not predict memorization, contrary to the susceptibility $\zeta$, which is strongly correlated with the accuracy on the noisy subset (Pearson correlation between Train ACC Noisy and susceptibility $\zeta$ is $\rho = 0.636$).

**Theoretical Results** To analyze the convergence of different models on some randomly labeled set, we rely on recent results obtained by modelling the training process of wide neural networks by neural tangent kernels (NTK) (Du et al., 2018; Jacot et al., 2018; Allen-Zhu et al., 2018; Arora et al., 2019; Bietti & Mairal, 2019; Lee et al., 2019). In particular, the analysis in our paper is motivated by recent work on convergence analysis of neural networks (Du et al., 2018; Arora et al., 2019). Arora et al. (2019) perform fine-grained convergence analysis for wide two-layered neural networks using gradient descent. They study the convergence of models trained on different datasets (datasets with clean labels versus random labels). In Section 5, we study the convergence of different models trained on some randomly-labeled set. In particular, our models are obtained by training wide two-layered neural networks on datasets with varying label noise levels. As we highlight in the proofs, our work builds on previous art, especially (Du et al., 2018; Arora et al., 2019), yet we build on a non-trivial way and for a new problem statement.

**Robustness to Adversarial Examples** Deep neural networks are vulnerable to adversarial samples (Szegedy et al., 2013; Goodfellow et al., 2014; Akhtar & Mian, 2018): very small changes to the input image can fool even state-of-the-art neural networks. This is an important issue that needs to be addressed for security reasons. To this end, multiple studies towards generating adversarial examples and defending against them have emerged (Carlini & Wagner, 2017; Papernot et al., 2016b;a; Athalye et al., 2018). As a side topic related to the central theme of the paper, we examined the connection between models that are resistant to memorization of noisy labels and models that are robust to adversarial attacks. Figure 8 compares the memorization of these models trained on some noisy dataset, with their robustness to adversarial attacks. We do not observe any positive or negative correlation between the two. Some models are robust to adversarial attacks, but perform poorly when trained on datasets with noisy labels, and vice versa. This means that the observations made in this paper are orthogonal to the ongoing discussion regarding the trade-off between robustness to adversarial samples and test accuracy (Tsipras et al., 2018; Stutz et al., 2019; Yang et al., 2020; Pedraza et al., 2021; Zhang et al., 2019b; Yin et al., 2019).

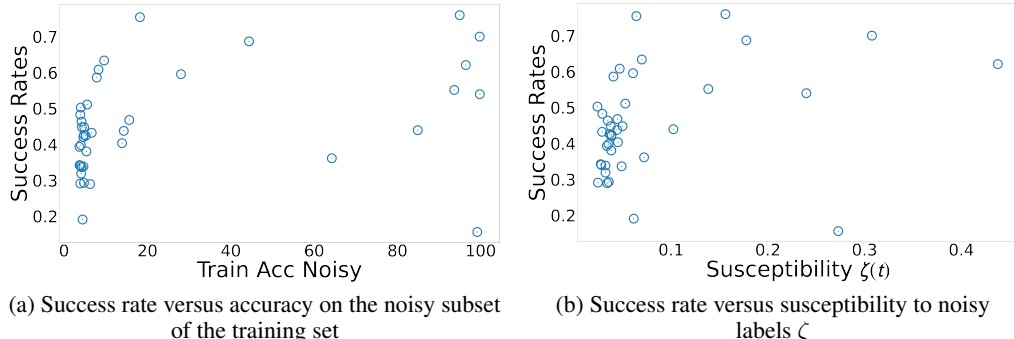

(a) Success rate versus accuracy on the noisy subset
of the training set

(b) Success rate versus susceptibility to noisy
labels $\zeta$

Figure 8: SimBa (Guo et al., 2019) adversarial attack success rate versus accuracy on the noisy subset of the training set for networks trained on CIFAR-10 with $50\%$ label noise. To evaluate robustness of different models with respect to adversarial attacks, we compare the success rate of the adversarial attack SimBa proposed in (Guo et al., 2019) after 1000 iterations. We observe no correlation between the success rate and neither accuracy on the noisy subset nor susceptibility $\zeta$.

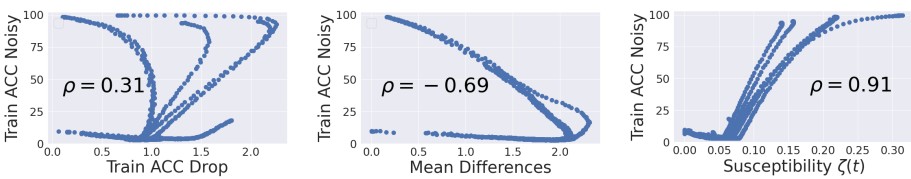

Figure 9: **Left:** Accuracy on the noisy subset of the training set, versus the training accuracy drop presented in (Zhang et al., 2019c) for the following neural network configurations trained on the CIFAR-10 dataset with $50\%$ label noise: a 5-layer convolutional neural network, DenseNet, EfficientNet, MobileNet, MobileNetV2, RegNet, ResNet, ResNeXt, SENet, and ShuffleNetV2. As proposed in (Zhang et al., 2019c), to compute the training accuracy drop, we create a new dataset from the available noisy training set, by replacing $25\%$ of its labels with random labels. We then train these networks on the new dataset, and then compute the difference in the training accuracy of the two setups, divided by 25 (the level of label noise that was injected). This is reported above as Train ACC Drop. (Zhang et al., 2019c) state that desirable networks should have a large drop; therefore there should ideally be a *negative* correlation between the training accuracy drop and the accuracy on the noisy subset. However, we observe a slight positive correlation. Therefore, it is difficult to use the results of Zhang et al. (2019c) to track memorization, i.e., training accuracy on the noisy subset. **Middle:** Accuracy on the noisy subset of the training set, versus the mean difference between the distributions obtained by applying GMM on the training losses, which is proposed in Lu & He (2022) as an early stopping criterion. We observe a negative correlation between the two, and hence conclude that this metric cannot be used to compare memorization of different settings at different stages of training. **Right:** As opposed to the other two metrics, we observe a strong positive correlation between our metric susceptibility $\zeta$ and memorization.

# B  EXPERIMENTAL SETUP

**Generating Noisy-labeled Datasets** We modify original datasets similarly to Chatterjee (2020); for a fraction of samples denoted by the label noise level (LNL), we replace the labels with independent random variables drawn uniformly from $\{1, \cdots, c\}$ for a dataset with $c$ number of classes. On average $(1 - \text{LNL}) + (\text{LNL}) \cdot 1/c$ of the samples still have their correct labels. Note that LNL for $\widetilde{S}$ is 1.

**Experiments of Figures 2, 12, 13 and 14** The models at epoch 0 are pre-trained models on either the clean or noisy versions of the CIFAR-10 dataset for 200 epochs using SGD with learning rate 0.1, momentum 0.9, weight decay $5 \cdot 10^{-4}$, and `cosineannealing` learning rate schedule with `T max` = 200. The new sample is a sample drawn from the training set, and its label is randomly assigned to some class different from the correct class label. In Figure 13, the new sample is an unseen sample drawn from the test set.

**CIFAR-10[4] Experiments** The models are trained for 200 epochs on the cross-entropy objective function using SGD with weight decay $5 \cdot 10^{-4}$ and batch size 128. The neural network architecture options are: cnn (a simple 5-layer convolutional neural network), DenseNet (Huang et al., 2017), EfficientNet (Tan & Le, 2019) (with `scale=0.5, 0.75, 1, 1.25, 1.5`), GoogLeNet (Szegedy et al., 2015), MobileNet (Howard et al., 2017) (with `scale=0.5, 0.75, 1, 1.25, 1.5`), ResNet (He et al., 2016a), MobileNetV2 (Sandler et al., 2018) (with `scale=0.5, 0.75, 1, 1.25, 1.5`), Preact ResNet (He et al., 2016b), RegNet (Radosavovic et al., 2020), ResNeXt (Xie et al., 2017), SENet (Hu et al., 2018), ShuffleNetV2 (Ma et al., 2018) (with `scale=0.5, 1, 1.5, 2`), DLA (Yu et al., 2018), and VGG (Simonyan & Zisserman, 2014). The learning rate value options are: 0.001, 0.005, 0.01, 0.05, 0.1, 0.5. The learning rate schedule options are: `cosineannealing` with `T max` 200, `cosineannealing` with `T max` 100, `cosineannealing` with `T max` 50, and no learning rate schedule. Momentum value options are 0.9, 0. In addition, we include experiments with Mixup (Zhang et al., 2017), active passive losses: normalized cross entropy with reverse cross entropy (NCE+RCE) (Ma et al., 2020), active passive losses: normalized focal loss with reverse cross entropy (NFL+RCE) (Ma et al., 2020), and robust early learning (Xia et al., 2021) regularizers.

**CIFAR-100 Experiments** The models are trained for 200 epochs on the cross-entropy objective function using SGD with learning rate 0.1, weight decay $5 \cdot 10^{-4}$, momentum 0.9, learning rate schedule `cosineannealing` with `T max` 200 and batch size 128. The neural network architecture options are: cnn, DenseNet, EfficientNet (with `scale=0.5, 0.75, 1, 1.25, 1.5`), GoogLeNet, MobileNet (with `scale=0.5, 0.75, 1, 1.25, 1.5`), ResNet, MobileNetV2 (with `scale=0.5, 0.75, 1, 1.25, 1.5`), RegNet, ResNeXt, ShuffleNetV2 (with `scale=0.5, 1, 1.5, 2`), DLA, and VGG. In addition, we include experiments with Mixup, active passive losses: NCE+RCE, active passive losses: NFL+RCE, and robust early learning regularizers.

**SVHN (Netzer et al., 2011) Experiments** The models are trained for 200 epochs on the cross-entropy objective function with learning rate 0.1 and batch size 128. The optimizer choices are SGD with weight decay $5 \cdot 10^{-4}$, and momentum 0.9, and Adam optimizers. The learning rate schedule options for the SGD experiments are: `cosineannealing` with `T max` 200, and `exponential`. The neural network architecture options are: DenseNet, EfficientNet (with `scale=0.25, 0.5, 0.75, 1`), GoogLeNet (with `scale= 0.25, 1, 1.25`), MobileNet (with `scale= 1, 1.25, 1.5, 1.75`), MobileNetV2 (with `scale= 1, 1.25, 1.5, 1.75`), ResNet (with `scale=0.25, 0.5, 1, 1.25, 1.5, 1.75`), ResNeXt, SENet (with `scale=0.25, 0.5, 0.75, 1`), ShuffleNetV2 (with `scale=0.25, 0.5, 0.75, 1`), DLA (with `scale=0.25, 0.5, 0.75, 1`), and VGG (with `scale=0.25, 0.5, 0.75, 1, 1.25, 1.5, 1.75`).

**MNIST[5] and Fashion-MNIST (Xiao et al., 2017) Experiments** The models are trained for 200 epochs on the cross-entropy objective function with batch size 128, using SGD with learning rate 0.1, weight decay $5 \cdot 10^{-4}$, and momentum 0.9. The learning rate

---

[4]`https://www.cs.toronto.edu/~kriz/cifar.html`
[5]`http://yann.lecun.com/exdb/mnist/`

schedule options are: `cosineannealing` with `T max 200`, and `exponential`. The neural network architecture options are: AlexNet (Krizhevsky et al., 2012) (with `scale=0.25, 0.5, 0.75, 1`), ResNet18 (with `scale=0.25, 0.5, 0.75, 1`), ResNet34 (with `scale=0.25, 0.5, 0.75, 1`), ResNet50 (with `scale=0.25, 0.5, 0.75, 1`), VGG11 (with `scale=0.25, 0.5, 0.75, 1`), VGG13 (with `scale=0.25, 0.5, 0.75, 1`), VGG16 (with `scale=0.25, 0.5, 0.75, 1`), VGG19 (with `scale=0.25, 0.5, 0.75, 1`).

**Tiny ImageNet (Le & Yang, 2015) Experiments** The models are trained for 200 epochs on the cross-entropy objective function with batch size 128, using SGD with weight decay $5 \cdot 10^{-4}$, and momentum 0.9. The learning rate schedule options are: `cosineannealing` with `T max 200`, `exponential`, and no learning rate schedule. The learning rate options are: $0.01, 0.05, 0.1, 0.5$. The neural network architecture options are: AlexNet, DenseNet, MobileNetV2, ResNet, SqueezeNet (Iandola et al., 2016), and VGG.

**Clothing-1M (Xiao et al., 2015) Experiments** The models are trained for 20 epochs on the cross-entropy objective function with batch size 128 using SGD with weight decay $5 \cdot 10^{-4}$, and momentum 0.9. The learning rate schedule options are: `cosineannealing` with `T max 20`, and `exponential`. The learning rate options are: $0.01, 0.005, 0.001$. The neural network architecture options are: AlexNet, ResNet18, ResNet 34, ResNeXt, and VGG. Note that because this dataset has random labels we do not introduce synthetic label noise to the labels.

**Animal-10N (Song et al., 2019) Experiments** The models are trained for 50 epochs on the cross-entropy objective function with batch size 128 using SGD with weight decay $5 \cdot 10^{-4}$, and momentum 0.9. The learning rate schedule options are: `cosineannealing` with `T max 50`, and `exponential`. The learning rate options are: $0.001, 0.005, 0.01$. The neural network architecture options are: ResNet18, ResNet34, SqueezeNet, AlexNet and VGG. Note that because this dataset has random labels we do not introduce synthetic label noise to the labels.

**CIFAR-10N (Wei et al., 2022) Experiments** On these experiments, we work with the aggregate version of the label set which has label noise level of $9\%$ (CIFAR-10N-aggregate in Table 1 of Wei et al. (2022)). The models are trained for 200 epochs on the cross-entropy objective function with batch size 128 using SGD with weight decay $5 \cdot 10^{-4}$, and momentum 0.9 with `cosineannealing` learning rate schedule with `T max 200`. The learning rate options are: $0.1, 0.05$. The neural network architecture options are: cnn, EfficientNet, MobileNet, ResNet, MobileNetV2, RegNet, ResNeXt, ShuffleNetV2, SENet, DLA and VGG.

Each of our experiments take few hours to run on a single Nvidia Titan X Maxwell GPU.

## C COMPARISON WITH BASELINES

**Comparison to a Noisy Validation Set** Following the notations from Chen et al. (2021), let the clean and noisy data distributions be denoted by $D$ and $\tilde{D}$, respectively, the classifiers by $h$, and the classification accuracy by $A$. Suppose that the optimal classifier in terms of accuracy on the clean data distribution is $h^*$, i.e., $h^* = \arg\max_h A_D(h)$. (Chen et al., 2021) states that under certain assumptions on the label noise, the accuracy on the noisy data distribution is also maximized by $h^*$, that is $h^* = \arg\max_h A_{\tilde{D}}(h)$. However, these results do not allow us to compare any two classifiers $h_1$ and $h_2$, because we cannot conclude from the results in (Chen et al., 2021) that if $A_D(h_1) > A_D(h_2)$, then $A_{\tilde{D}}(h_1) > A_{\tilde{D}}(h_2)$. Moreover, it is important to note that these results hold when the accuracy is computed on unlimited dataset sizes. As stated in (Lam & Stork, 2003), the number of noisy validation samples that are equivalent to a single clean validation sample depends on the label noise level and on the true error rate of the model, which are both unknown in practice. Nevertheless, below we thoroughly compare our approach with having a noisy validation set.

As discussed in Section 4, the susceptibility $\zeta$ together with the training accuracy are able to select models with a high test accuracy. Another approach to select models is to use a subset of the available noisy dataset as a held-out validation set. Table 1 provides the correlation values between the test accuracy on the clean test set and the validation accuracy computed on noisy validation sets with varying sizes. On the one hand, we observe that the correlation between the validation accuracy and the test accuracy is very low for small sizes of the noisy validation set. On the other hand, in the same

| Size of the set | Train Acc | Val Acc | Our approach |
|:---:|:---:|:---:|:---:|
| 10 | | 0.244 | **0.792** |
| 128 | | 0.458 | **0.890** |
| 256 | | 0.707 | **0.878** |
| 512 | | 0.799 | **0.876** |
| 1024 | 0.513 | 0.830 | **0.877** |
| 2048 | | 0.902 | **0.917** |
| 4096 | | 0.886 | **0.912** |
| 8192 | | **0.940** | 0.923 |
| 10000 | | **0.956** | 0.914 |

Table 1: The Kendall $\tau$ correlation between each metric and the test accuracy for CNN, ResNet, EfficientNet, and MobileNet trained on CIFAR-10 with $50\%$ label noise, where the validation accuracy (Val Acc) on a held-out subset of the data and the susceptibility $\zeta$ use a set with the size indicated in each row. We observe that our approach results in a much higher correlation compared to using a noisy validation set. Furthermore, our approach is less sensitive to the size of the held-out set, compared to using a noisy validation set. In particular, to reach the same correlation value, our approach with set size $= 10$ is equivalent to using a noisy validation set with size $= 512$ (highlighted in red). Also, our approach with set size $= 128$ is almost equivalent to using a noisy validation set with size $= 1024$ (highlighted in blue). Hence, to use a noisy validation set, one requires around ten-fold the amount of held-out data.

table, we observe that if the same size is used to compute $\zeta$, our approach provides a high correlation to the test accuracy, even for very small sizes of the held-out set.

In our approach, we first filter out models for which the value of $\zeta$ exceeds a threshold. We set the threshold so that around half of the models remain after this filtration. We then report the correlation between the training and test accuracies among the remaining models. As a sanity check, we doubled checked that, with this filtration, the model with the highest test accuracy was not filtered out.

Moreover, in Figure 10, we report the advantage of using our approach compared to using a noisy validation set for various values of the dataset label noise level (LNL) and the size of the set that computes the validation accuracy and susceptibility $\zeta$. We observe that the lower the size of the validation set, and the higher the LNL, the more advantageous our approach is. Note also that for high set sizes and low LNLs, our approach produces comparable results to using a noisy validation set.

**Comparison to Label Noise Detection Approaches** Another line of work is studying methods that detect whether a label assigned to a given sample is correct or not (Zhu et al., 2021; Song et al., 2020b; Pleiss et al., 2020; Pulastya et al., 2021). Such methods can estimate the clean and noisy subsets. Then, by tracking the training accuracy on the clean and noisy subsets, similar to what is done in Figure 5, they can select models that are located in Region 1, i.e., that have a low estimated accuracy on the noisy subset and a high estimated accuracy on the clean subset. In Figure 11, we compare the average test accuracy of models selected by our approach with those selected by such subset-selection methods. Let $X$ be the accuracy of the detection of the correct/incorrect label of a sample by the subset selection benchmark: if $X = 100\%$, then the method has full access to the ground truth label for each label, if $X = 90\%$ the method correctly detects $90\%$ of the labels. We observe a clear advantage of our method for $X$ up to $96\%$, and comparable performance for $X > 96\%$.

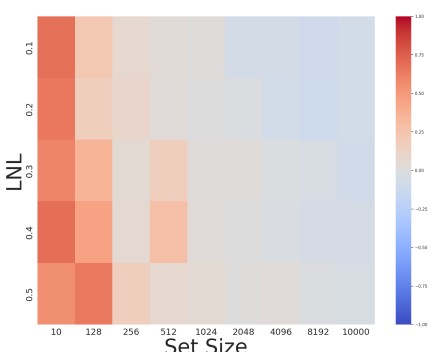

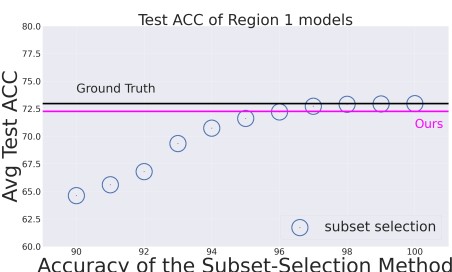

Figure 10: The Kendall $\tau$ correlation between our approach and the test accuracy minus the Kendall $\tau$ correlation between validation accuracy (computed on a noisy set) and the test accuracy for various label noise levels (LNL) and set sizes. We observe the advantage of our approach to using a noisy validation set particularly for high LNLs and low set sizes. For other combinations, we also observe comparable results (the correlation difference is very close to zero). Note that the last row can be recovered from the difference in correlation values of Table 1.

Figure 11: Comparison between the average test accuracy obtained using our approach and the average test accuracy obtained using a method that detects correctly-labeled samples within the training set from the incorrectly-labeled ones with $X\%$ accuracy (the x-axis). The results are obtained from training {CNN, ResNet, EfficientNet, MobileNet}$\times${without regularization, +Mixup, +NCERCE, + NFLRCE, +robust early learning} on CIFAR-10 with $50\%$ label noise. We observe that to have the same performance as our approach, such methods require a very high accuracy (above $96\%$), and even with higher accuracies, our approach gives comparable results.

# D    ADDITIONAL EXPERIMENTS FOR SECTION 2

In this section, we provide additional experiments for the observation presented in Section 2. In Figure 12, we observe that networks with a high test accuracy are resistant to memorizing a new incorrectly-labeled sample. On the other hand, in Figure 13, we observe that networks with a high test accuracy tend to fit a new correctly-labeled sample faster.

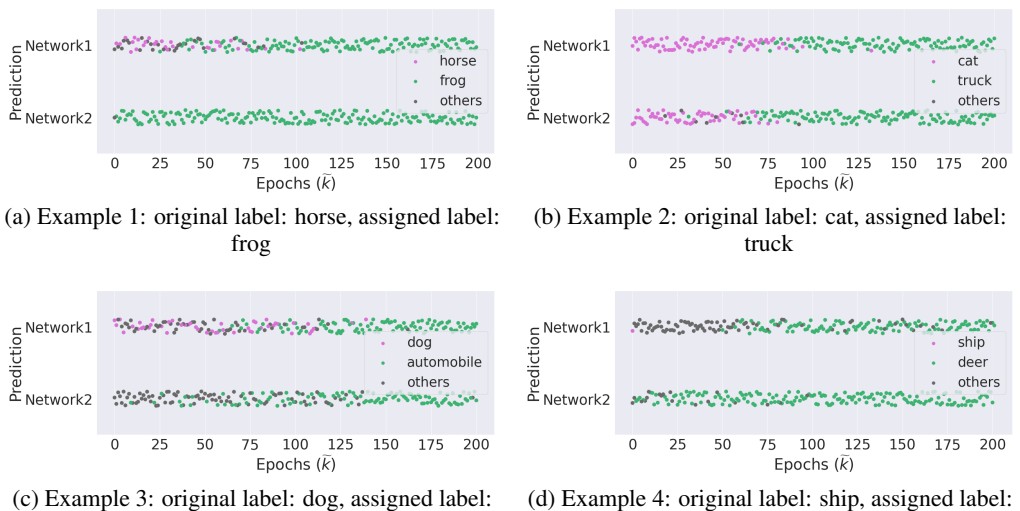

(a) Example 1: original label: horse, assigned label: frog

(b) Example 2: original label: cat, assigned label: truck

(c) Example 3: original label: dog, assigned label: automobile

(d) Example 4: original label: ship, assigned label: deer

Figure 12: The evolution of output prediction of two networks that are trained on a single randomly labeled sample. In all sub-figures, Network 1 has a higher test accuracy compared to Network 2, and we observe it is less resistance to memorization of the single incorrectly-labeled sample. **Example 1:** Network 1 is a ResNeXt trained on CIFAR-10 dataset with $50\%$ random labels and has test accuracy of $58.85\%$. Network 2 is a ResNeXt that is not pre-trained and has test accuracy of $9.74\%$. **Example 2:** Network 1 is a SENet trained on CIFAR-10 dataset with original labels and has test accuracy of $95.35\%$. Network 2 is a SENet that is trained on CIFAR-10 with $50\%$ label noise and has test accuracy of $56.38\%$. **Example 3:** Network 1 is a RegNet trained on CIFAR-10 dataset with original labels and has test accuracy of $95.28\%$. Network 2 is a RegNet that is trained on CIFAR-10 with $50\%$ label noise and has test accuracy of $55.36\%$. **Example 4:** Network 1 is a MobileNet trained on CIFAR-10 dataset with original labels and has test accuracy of $90.56\%$. Network 2 is a MobileNet that is trained on CIFAR-10 with $50\%$ label noise and has test accuracy of $82.76\%$.

Moreover, we study the effect of calibration on the observations of Figures 2 and 12. A poor calibration of a model may affect the confidence in its predictions, which in turn might affect the susceptibility/resistance to new samples. Therefore, in Figure 14, we compare models that have almost the same calibration value. More precisely, Network 1 is trained on the clean dataset, and Network 2 (calibrated) is a calibrated version of the model that is trained on the noisy dataset using the Temperature scaling approach (Guo et al., 2017). We observe that even with the same calibration level, the model with a higher test accuracy is more resistant to memorizing a new incorrectly-labeled sample.

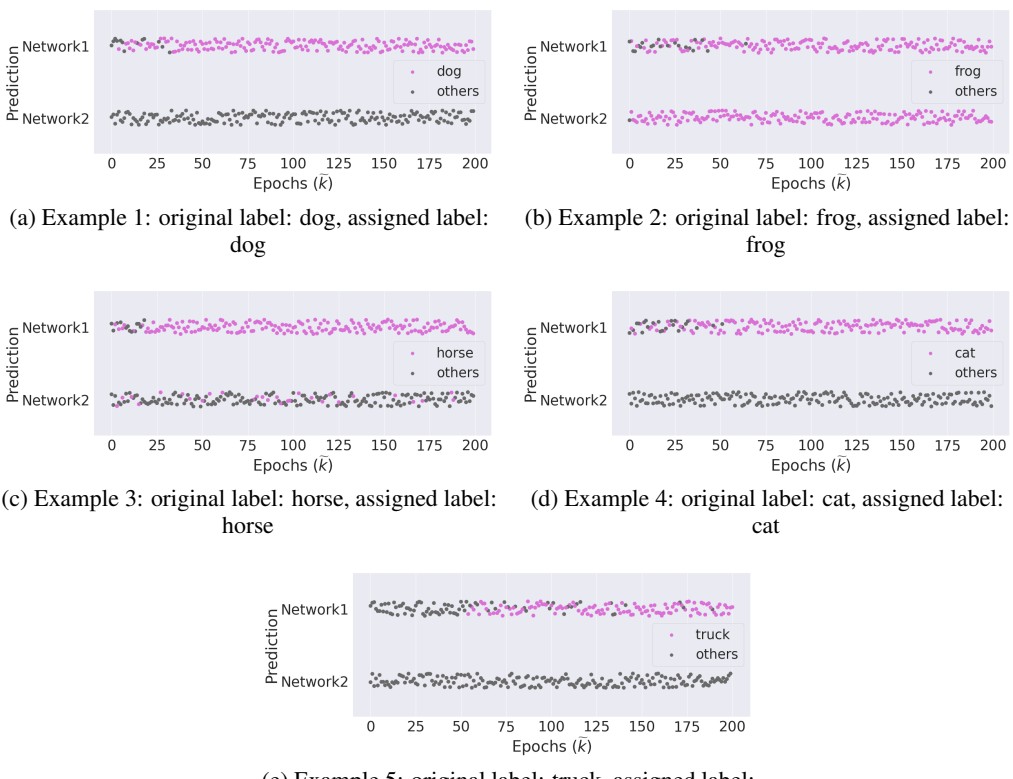

(a) Example 1: original label: dog, assigned label: dog

(b) Example 2: original label: frog, assigned label: frog

(c) Example 3: original label: horse, assigned label: horse

(d) Example 4: original label: cat, assigned label: cat

(e) Example 5: original label: truck, assigned label: truck

Figure 13: The evolution of output prediction of two networks that are trained on a single unseen correctly-labeled sample. In all sub-figures, Network 1 has a higher test accuracy compared to Network 2. We observe that give a new correctly-labeled sample Network 2 learns it later, unlike our observation in Figure 2 for a new incorrectly-labeled sample. **Example 1:** Network 1 is a GoogLeNet trained on CIFAR-10 dataset with clean labels and has test accuracy of $95.36\%$. Network 2 is a GoogLeNet trained on CIFAR-10 dataset with $50\%$ label noise level and has test accuracy of $58.35\%$. **Example 2:** Network 1 is a ResNeXt trained on CIFAR-10 dataset with $50\%$ random labels and has test accuracy of $58.85\%$. Network 2 is a ResNeXt that is not pre-trained and has test accuracy of $9.74\%$. **Example 3:** Network 1 is a SENet trained on CIFAR-10 dataset with original labels and has test accuracy of $95.35\%$. Network 2 is a SENet that is trained on CIFAR-10 with $50\%$ label noise and has test accuracy of $56.38\%$. **Example 4:** Network 1 is a RegNet trained on CIFAR-10 dataset with original labels and has test accuracy of $95.28\%$. Network 2 is a RegNet that is trained on CIFAR-10 with $50\%$ label noise and has test accuracy of $55.36\%$. **Example 5:** Network 1 is a MobileNet trained on CIFAR-10 dataset with original labels and has test accuracy of $90.56\%$. Network 2 is a MobileNet that is trained on CIFAR-10 with $50\%$ label noise and has test accuracy of $82.76\%$.

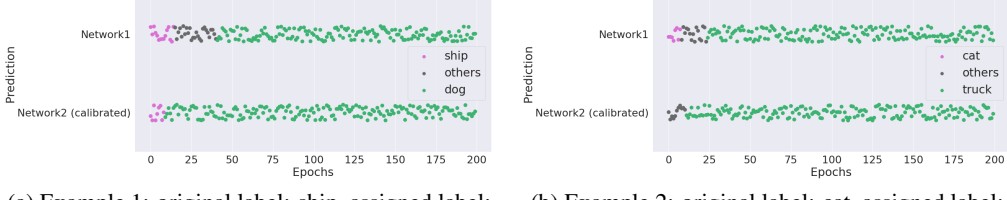

(a) Example 1: original label: ship, assigned label: dog

(b) Example 2: original label: cat, assigned label: truck

Figure 14: The evolution of output prediction of networks that are trained on a single randomly labeled sample. In all sub-figures, Network 1 (trained on the clean dataset) has a higher test accuracy than Network 2 (trained on the noisy dataset), and we observe it is less resistant to memorization of the single incorrectly-labeled sample. Furthermore, we have ensured using the temperature scaling method (Guo et al., 2017) that the two models have the same calibration (ECE) value. **Example 1:** Network 1 is a GoogleNet trained on CIFAR-10 dataset with original labels and has test accuracy of $95.36\%$. Network 2 is a GoogleNet that is trained on CIFAR-10 with $50\%$ label noise and has test accuracy of $58.35\%$. **Example 2:** Network 1 is a RegNet trained on CIFAR-10 dataset with original labels and has test accuracy of $95.28\%$. Network 2 is a RegNet that is trained on CIFAR-10 with $50\%$ label noise and has test accuracy of $55.36\%$.

# E ADDITIONAL EXPERIMENTS FOR SECTION 4

In this section, we provide additional experiments for our main results for MNIST, Fashion-MNIST, SVHN, CIFAR-100, and Tiny Imagenet datasets. Later in the section, we provide additional experiments on the Clothing-1M, Animal-10N and CIFAR-10N datasets, where as stated in Appendix B are datasets with real-world label noise.

In Figure 15, we observe that for networks trained on the noisy MNIST datasets, models that are resistant to memorization and trainable have on average more than $20\%$ higher test accuracy compared to models that are trainable but not resistant (similar results for other datasets are observed in Figures 17, 19, 22, and 25). Furthermore, without access to the ground-truth, the models with a high (respectively, low) accuracy on the clean (resp., noisy) subsets are recovered using susceptibility $\zeta$ as shown in Figures 20 and 23. Moreover, in Figure 16, we observe that by selecting models with a low value of $\zeta(t)$ the correlation between training accuracy and test accuracy drastically increases from $-0.766$ to $0.863$, which shows the effectiveness of the susceptibility metric $\zeta(t)$ (similar results for other datasets are observed in Figures 18, 21, 24, and 26).

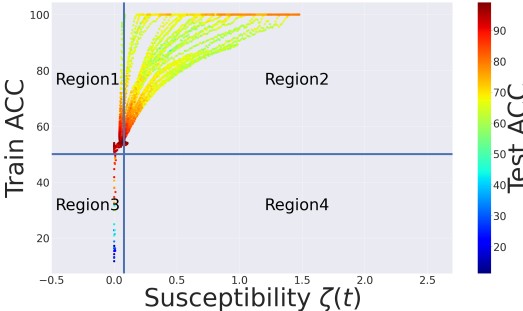

Figure 15: Using susceptibility $\zeta(t)$ and training accuracy, we can obtain 4 different regions for models trained on MNIST with $50\%$ label noise (details in Appendix B). **Region 1**: Trainable and resistant, with average test accuracy of $95.38\%$. **Region 2**: Trainable and but not resistant, with average test accuracy of $72.65\%$. **Region 3**: Not trainable but resistant, with average test accuracy of $47.69\%$. **Region 4**: Neither trainable nor resistant.

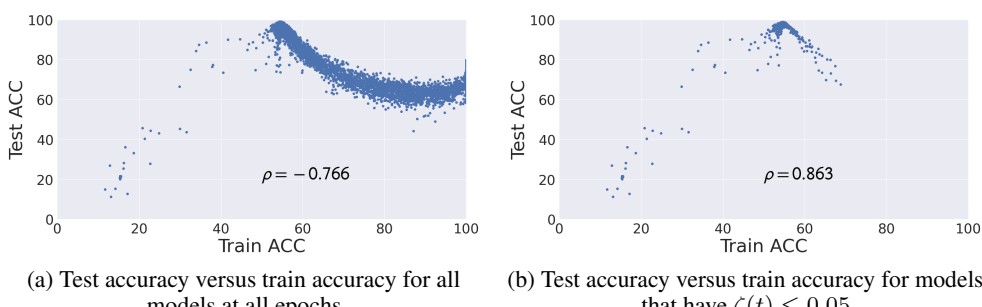

(a) Test accuracy versus train accuracy for all models at all epochs

(b) Test accuracy versus train accuracy for models that have $\zeta(t) \leq 0.05$

Figure 16: For models trained on MNIST with $50\%$ label noise (details in Appendix B), the correlation between training accuracy and test accuracy increases a lot by removing models based on the susceptibility metric $\zeta(t)$.

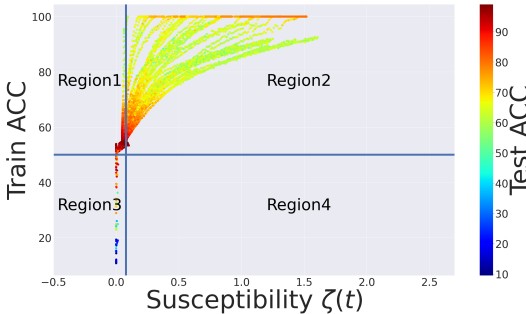

Figure 17: Using susceptibility $\zeta(t)$ and training accuracy we can obtain 4 different regions for models trained on Fashion-MNIST with $50\%$ label noise (details in Appendix B). **Region 1**: Trainable and resistant, with average test accuracy of $95.82\%$. **Region 2**: Trainable and but not resistant, with average test accuracy of $72.04\%$. **Region 3**: not trainable but resistant, with average test accuracy of $52.68\%$. **Region 4**: Neither trainable nor resistant.

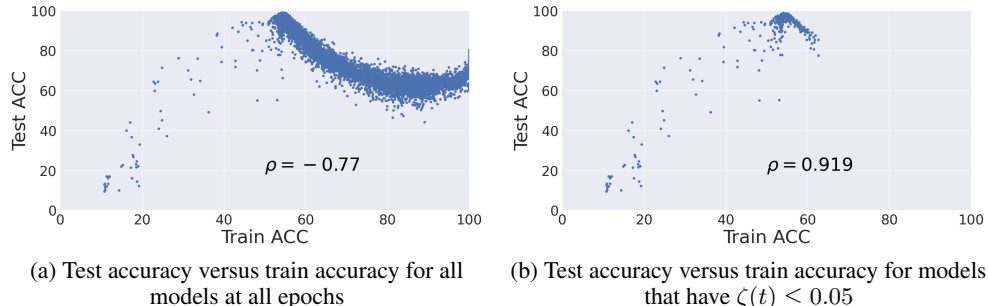

(a) Test accuracy versus train accuracy for all models at all epochs

(b) Test accuracy versus train accuracy for models that have $\zeta(t) \leq 0.05$

Figure 18: For models trained on Fashion-MNIST with $50\%$ label noise (details in Appendix B), the correlation between training accuracy and test accuracy increases a lot by removing models based on the susceptibility metric $\zeta(t)$.

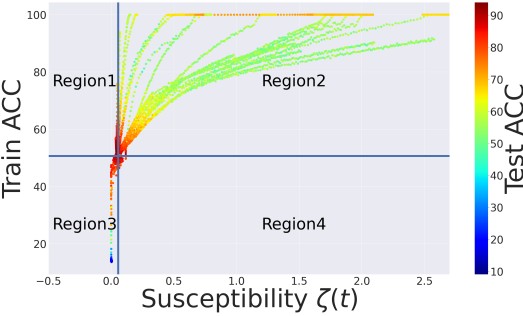

Figure 19: Using susceptibility $\zeta(t)$ and training accuracy we can obtain 4 different regions for models trained on SVHN with $50\%$ label noise (details in Appendix B). **Region 1**: Trainable and resistant, with average test accuracy of $88.64\%$. **Region 2**: Trainable and but not resistant, with average test accuracy of $66.34\%$. **Region 3**: Not trainable but resistant, with average test accuracy of $53.25\%$. **Region 4**: Neither trainable nor resistant, with average test accuracy of $85.17\%$.

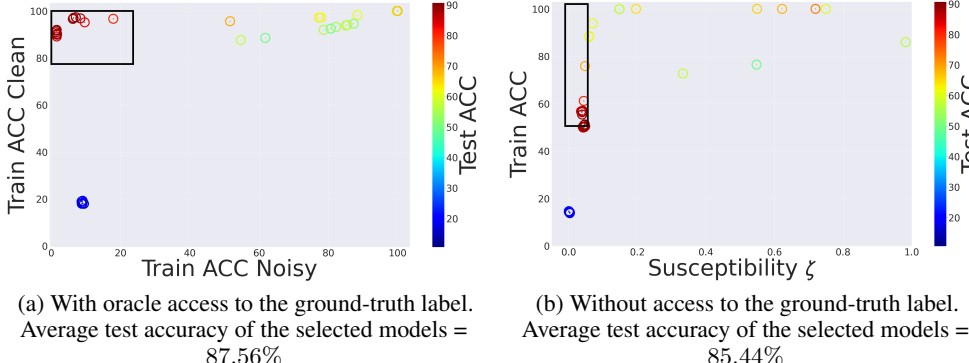

(a) With oracle access to the ground-truth label. Average test accuracy of the selected models = 87.56%

(b) Without access to the ground-truth label. Average test accuracy of the selected models = 85.44%

Figure 20: For models trained on SVHN with $50\%$ label noise (details in Appendix B), with the help of our susceptibility metric $\zeta(t)$ and the overall training accuracy, we can recover models with a high/low accuracy on the clean/noisy subset.

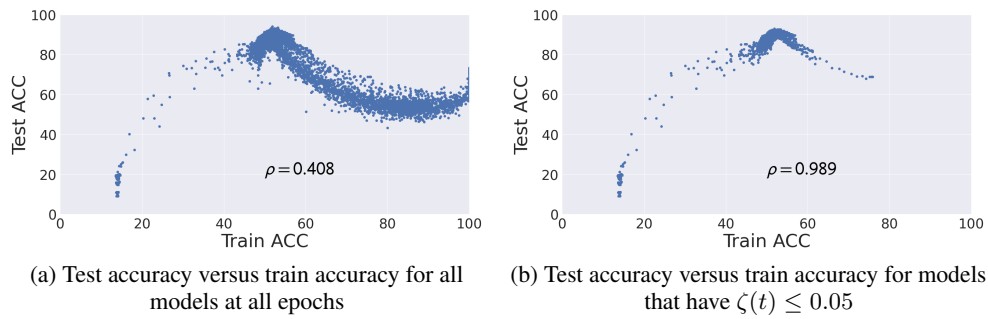

(a) Test accuracy versus train accuracy for all models at all epochs

(b) Test accuracy versus train accuracy for models that have $\zeta(t) \leq 0.05$

Figure 21: For models trained on SVHN with $50\%$ label noise (details in Appendix B), the correlation between training accuracy and test accuracy increases a lot by removing models based on the susceptibility metric $\zeta(t)$.

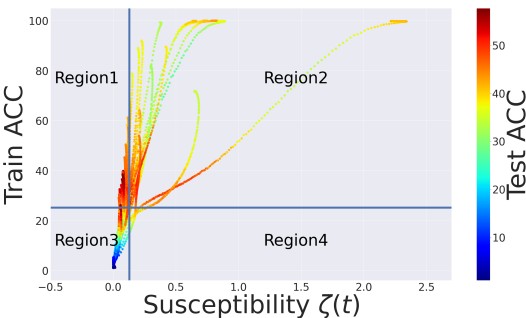

Figure 22: Using susceptibility $\zeta(t)$ and training accuracy we can obtain 4 different regions for models trained on CIFAR-100 with $50\%$ label noise (details in Appendix B). **Region 1**: Trainable and resistant, with average test accuracy of $47.09\%$. **Region 2**: Trainable and but not resistant, with average test accuracy of $40.96\%$. **Region 3**: Not trainable but resistant, with average test accuracy of $22.65\%$. **Region 4**: Neither trainable nor resistant, with average test accuracy of $39.07\%$.

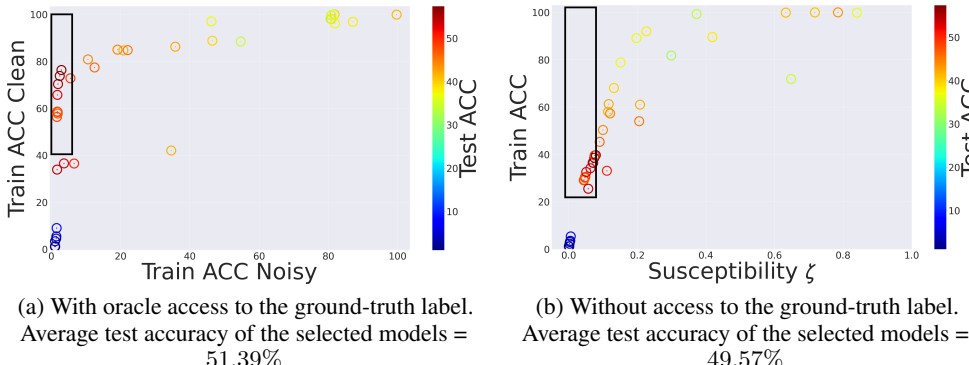

(a) With oracle access to the ground-truth label. Average test accuracy of the selected models = 51.39%

(b) Without access to the ground-truth label. Average test accuracy of the selected models = 49.57%

Figure 23: For models trained on CIFAR-100 with 50% label noise, with the help of our susceptibility metric $\zeta(t)$ and the overall training accuracy, we can recover models with a high/low accuracy on the clean/noisy subset.

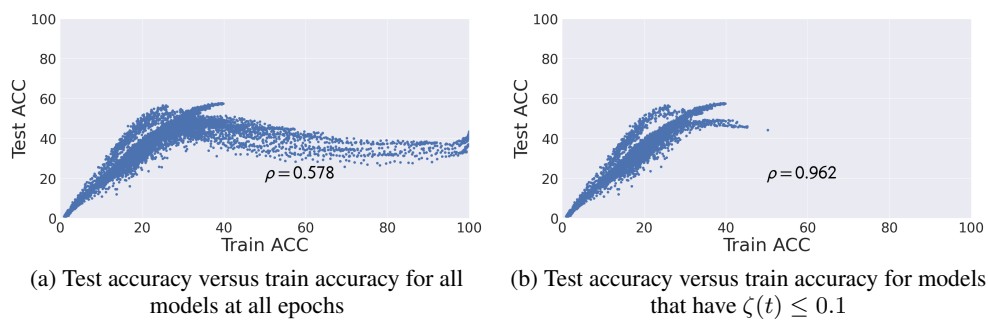

(a) Test accuracy versus train accuracy for all models at all epochs

(b) Test accuracy versus train accuracy for models that have $\zeta(t) \leq 0.1$

Figure 24: For models trained on CIFAR-100 with 50% label noise (details in Appendix B), the correlation between training accuracy and test accuracy increases a lot by removing models based on the susceptibility metric $\zeta(t)$.

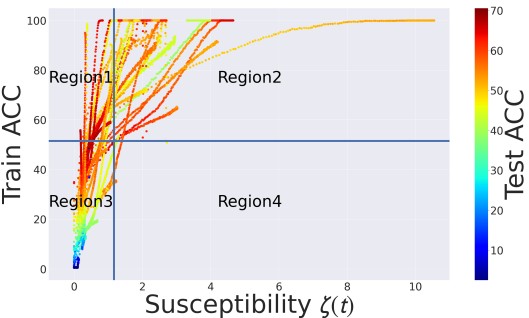

Figure 25: Using susceptibility $\zeta(t)$ and training accuracy we can obtain 4 different regions for models trained on Tiny Imagenet with 10% label noise (details in Appendix B). **Region 1**: Trainable and resistant, with average test accuracy of 57.51%. **Region 2**: Trainable and but not resistant, with average test accuracy of 53.25%. **Region 3**: Not trainable but resistant, with average test accuracy of 18.53%. **Region 4**: Neither trainable nor resistant, with average test accuracy of 53.26%.

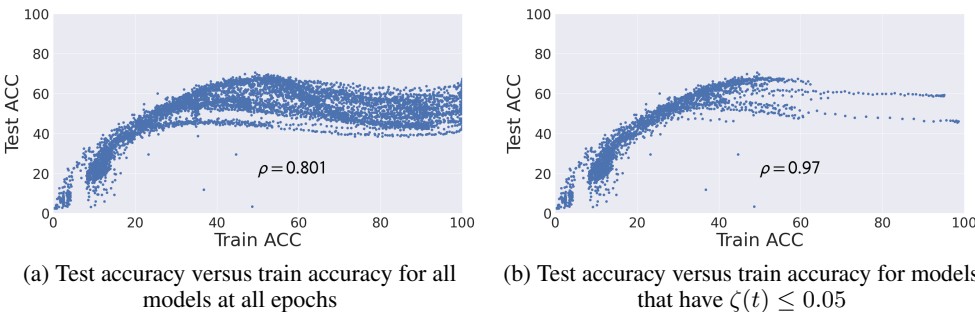

(a) Test accuracy versus train accuracy for all models at all epochs

(b) Test accuracy versus train accuracy for models that have $\zeta(t) \leq 0.05$

Figure 26: For models trained on Tiny Imagenet with $10\%$ label noise (details in Appendix B), the correlation between training accuracy and test accuracy increases by removing models based on the susceptibility metric $\zeta(t)$.

### E.1 WALK-THROUGH ON REAL-WORLD NOISY-LABALED DATASETS

To evaluate the performance of our approach on real-world datasets, we have conducted additional experiments on the Clothing-1M dataset (Xiao et al., 2015), which is a dataset with 1M images of clothes, on the Animal-10N dataset (Song et al., 2019), which is a dataset with 50k images of animals and on the CIFAR-10N dataset (Wei et al., 2022), which is the CIFAR-10 dataset with human-annotated noisy labels obtained from Amazon Mechanical Turk. In the Clothing-1M dataset, the images have been labeled from the texts that accompany them, hence there are both clean and noisy labels in the set, and in the Animal-10N dataset, the images have been gathered and labeled from search engines. In these datasets, some images have incorrect labels and the ground-truth labels in the training set are not available. Hence in our experiments we cannot explicitly track memorization as measured by the accuracy on the noisy subset of the training set.

We train different settings on these two datasets with various architectures (including ResNet, AlexNet and VGG) and varying hyper-parameters (refer to Appendix B for details). We compute the training accuracy and susceptibility $\zeta$ during the training process for each setting and visualize the results in Figure 27 below.

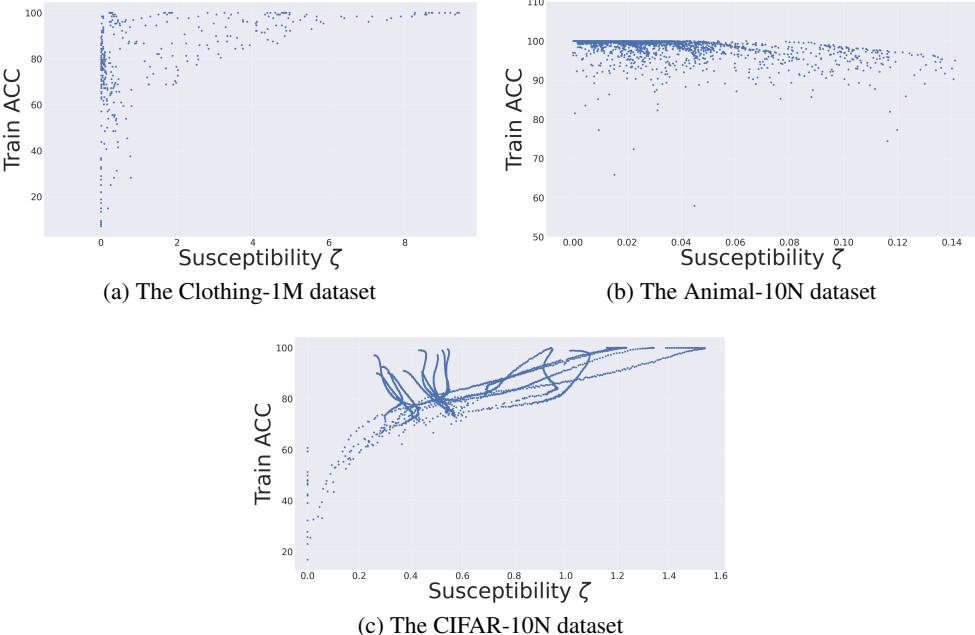

(a) The Clothing-1M dataset    (b) The Animal-10N dataset

(c) The CIFAR-10N dataset

Figure 27: Training accuracy and susceptibility $\zeta$ for various models trained on real-world noisy-labeled datasets.

We divide the models of Figure 27 into 4 regions, where the boundaries are set to the average value of the training accuracy (horizontal line) and the average value of susceptibility (vertical line) over the available models: Region 1: Models that are trainable and resistant to memorization, Region 2: Trainable and but not resistant, Region 3: Not trainable but resistant and Region 4: Neither trainable nor resistant. This is shown in Figure 28.

Our approach suggests selecting models in Region 1 (low susceptibility, high training accuracy).

In order to assess how our approach does in model-selection, we can reveal the test accuracy computed on a held-out clean test set in Figure 29. We observe that the average (± standard deviation) of the test accuracy of models in each region is as follows:

- *Clothing-1M dataset:* Region 1: **61.799**% ± 1.643; Region 2: 57.893% ± 3.562; Region 3: 51.250% ± 17.209; Region 4: 51.415% ± 9.709.

- *Animal-10N dataset:* Region 1: **96.371**% ± 1.649; Region 2: 92.508% ± 2.185; Region 3: 91.179% ± 6.601; Region 4: 89.352% ± 3.142.

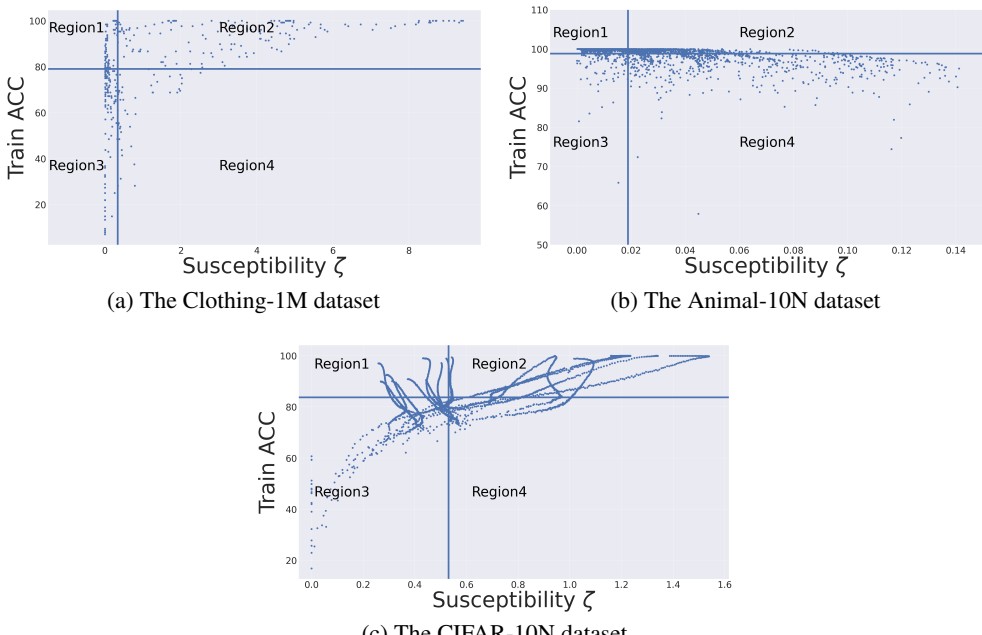

(a) The Clothing-1M dataset  (b) The Animal-10N dataset

(c) The CIFAR-10N dataset

Figure 28: Training accuracy and susceptibility $\zeta$ for various models trained on real-world noisy-labeled datasets. According to the average values of the training accuracy and susceptibility $\zeta$ the figure is divided into 4 regions.

- *CIFAR-10N dataset:* Region 1: **87.2**% ± 0.99; Region 2: $85.44\%$ ± 2.52; Region 3: 77.87% ± 8.15; Region 4: 78.45% ± 3.86.

We observe that using our approach we are able to select models with a very high test accuracy. In addition, the test accuracies of models in Region 1 have the least amount of standard deviation. Note that our susceptibility metric $\zeta$ does not use any information about the label noise level or the label noise type that is present in these datasets. Similarly to the rest of this paper, random labeling is used for computing $\zeta$. Interestingly, even though within the training sets of these datasets the label noise type is different than random labeling (label noise type is instance-dependent (Xia et al., 2020; Wei et al., 2022)), $\zeta$ is still successfully tracking memorization.

Therefore, our approach selects trainable models with low memorization even for datasets with real-world label noise. Observe that selecting models only on the basis of their training accuracy or only on the basis of their susceptibility fails: both are needed. It is interesting to note that in the Clothing-1M dataset, as the dataset is more complex, the range of the performance of different models varies and our approach is able to select "good" models from "bad" models. On the other hand, in the Animal-10N and CIFAR-10N datasets, as the datasets are easier to learn and the estimated label noise level is lower, most models are already performing rather well. Here, our approach is able to select the "best" models from "good" models.

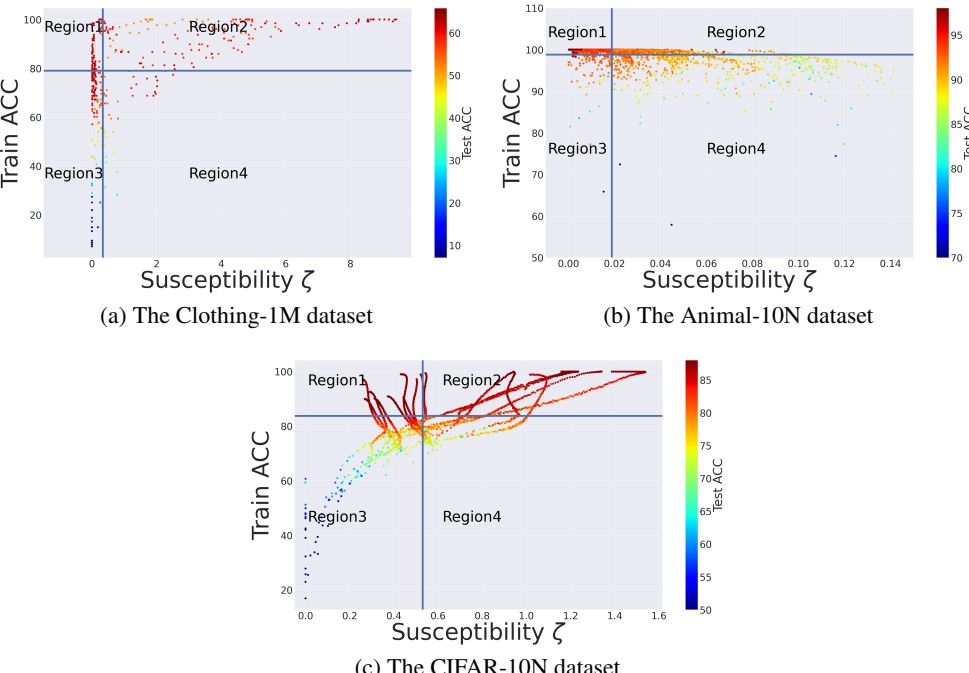

(a) The Clothing-1M dataset

(b) The Animal-10N dataset

(c) The CIFAR-10N dataset

Figure 29: Training accuracy and susceptibility $\zeta$ for various models trained on real-world noisy-labeled datasets. According to the average values of the training accuracy and susceptibility $\zeta$ the figure is divided into 4 regions. The test accuracy of the models is visualized using the color of each point and is only for illustration and is not used to find different regions.

# F    EXPERIMENTS RELATED TO SECTION 6

In this section, we provide some ablation studies that are discussed in Section 6.

In Figure 30, we observe that even among neural network architectures with a good resistance to memorization, susceptibility to noisy labels $\zeta(t)$ detects the most resistant model. We observe that the high correlation between $\zeta$ and memorization of the noisy subset is not limited to a specific learning rate schedule in Figure 31, or a label noise level in Figures 32 and 33. Moreover, in Figures 34 and 35, we observe that for datasets with label noise level of $10\%$, the susceptibility to noisy labels $\zeta$ and training accuracy still select models with a high test accuracy. The same consistency is observed in Figure 38 for models trained with asymmetric label noise.

In the paper, we choose $\widetilde{S}$ to be only a single mini-batch of randomly-labeled set for computational efficiency. But we also made sure that this does not harm the correlation between Train ACC Noisy and $\zeta(t)$. We analyze the effect of size of $\widetilde{S}$ in Figure 36 (left), which confirms that a single mini-batch is large enough to have a high correlation between Train ACC Noisy and $\zeta(t)$. Moreover, we observe the robustness of the susceptibility metric to the exact choice of the mini-batch in Figure 36 (right).

To better illustrate the match between Train ACC Noisy and $\zeta(t)$, we provide the overlaid curves in Figure 37. This figure clearly shows how using $\zeta$, one can detect/select checkpoints of the model with low memorization.

**A study on different thresholds used to select models** We would like to point out that if we can tune these thresholds (instead of using the average values of training accuracy and susceptibility over the available models), we can select models with even higher test accuracies than what is reported in our paper. For example, for models of Figure 5, by tuning these two thresholds one could reach a test accuracy of $79.15\%$ (instead of the reported $76\%$) as shown in Figure 39 (left). However, we want to remain in the practical setting where we do not have any access to a clean validation set for tuning. As a consequence, we must avoid any hyper-parameter tuning. And indeed, throughout our experiments, these thresholds are never tuned nor set manually to any extent. Among thresholds that can be computed without access to a clean validation set, we opted for the average values of susceptibility and training accuracy (over the available models) for simplicity. We empirically observe that this choice is robust and produces favorable results in various experimental settings. We could take other percentiles for the threshold, but they are more complex to obtain than simple averages, because they would then depend on the distribution among models. In Figure 39, we study various values of percentiles for these thresholds. We observe that depending on the available models and the given dataset, some other percentiles might give higher test accuracies than simply using the average values. These percentiles range however typically from 35 to 55 and are therefore not far from the mean, hence their benefit in increasing test accuracy appears small compared to the increased complexity to compute them or to rely on additional assumptions on the distribution of susceptibility and training accuracy. We observe in Figure 39 (right) that except for very extreme values of the thresholds (which basically select all models as resistant to memorization), the average test accuracy of models in Region 1 is much higher than the averge test accuracy of models in Region 2. Hence, our proposed model selection approach is robust to the choice of these thresholds.

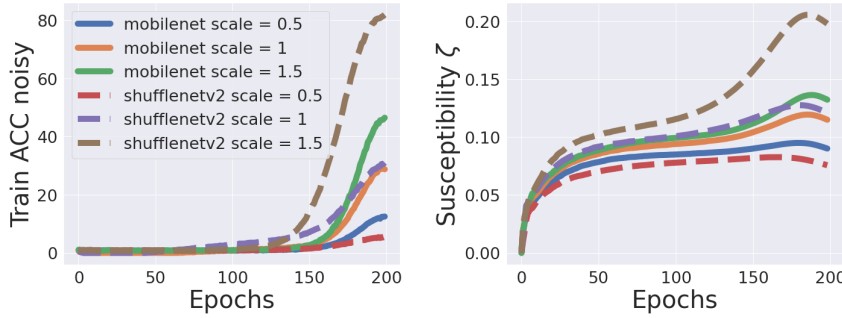

Figure 30: Accuracy on the noisy subset of the training set versus the susceptibility $\zeta(t)$ (Equation (2)) for MobileNet and ShuffleNetV2 configurations trained on CIFAR-100 with $50\%$ label noise. Pearson correlation between the Train ACC Noisy and susceptibility $\zeta$ is $\rho = 0.749$. `Scale` is a hyperparameter that proportionally scales the number of hidden units and number of channels in the neural network configuration.

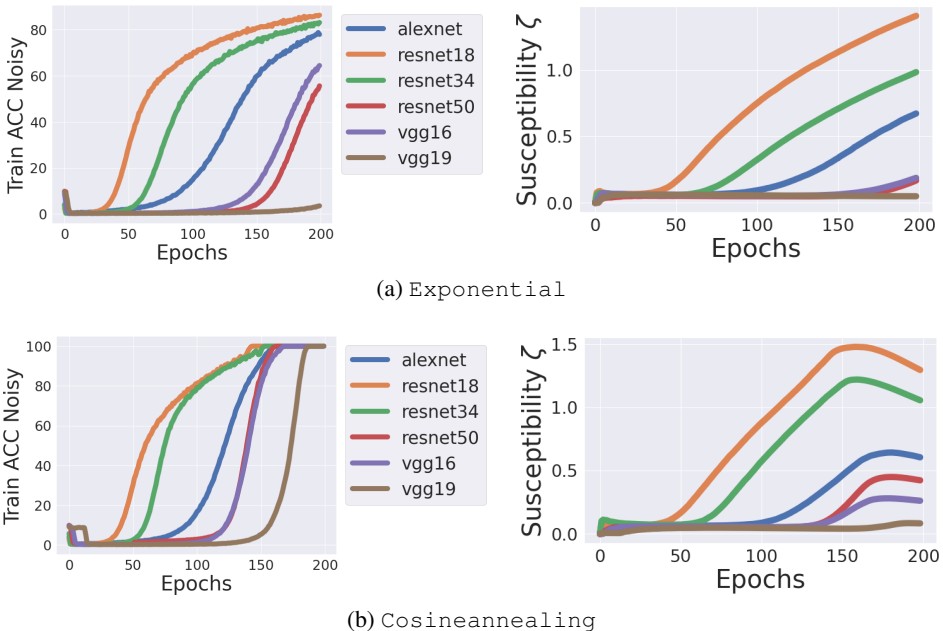

Figure 31: Accuracy on the noisy subset of the training set versus the susceptibility $\zeta(t)$ for networks trained on MNIST with $50\%$ label noise. On top and bottom we have models trained with `exponential` and `cosineannealing` learning rate schedulers, respectively. Pearson correlation between Train ACC Noisy and $\zeta$ for `exponential` and `cosineannealing` schedules are $\rho = 0.89$ and $\rho = 0.772$, respectively .

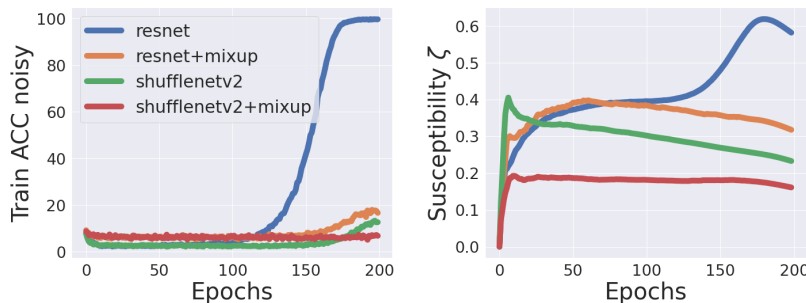

Figure 32: Accuracy on the noisy subset of the training set versus susceptibility to noisy labels $\zeta(t)$ for networks trained on CIFAR-10 with $10\%$ label noise. Pearson correlation between Train ACC Noisy and $\zeta$ is $\rho = 0.634$.

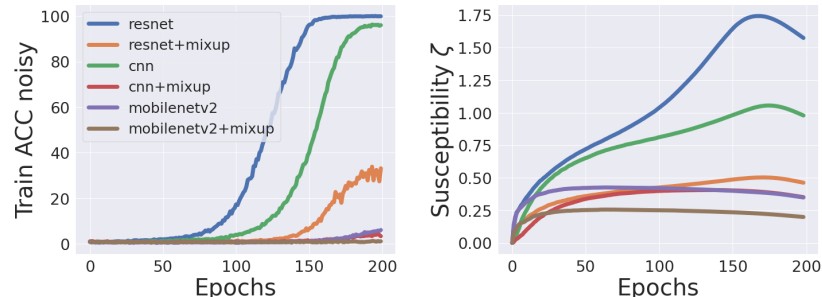

Figure 33: Accuracy on the noisy subset of the training set versus susceptibility to noisy labels $\zeta(t)$ for networks trained on CIFAR-100 with $10\%$ label noise. Pearson correlation between Train ACC Noisy and $\zeta$ is $\rho = 0.849$.

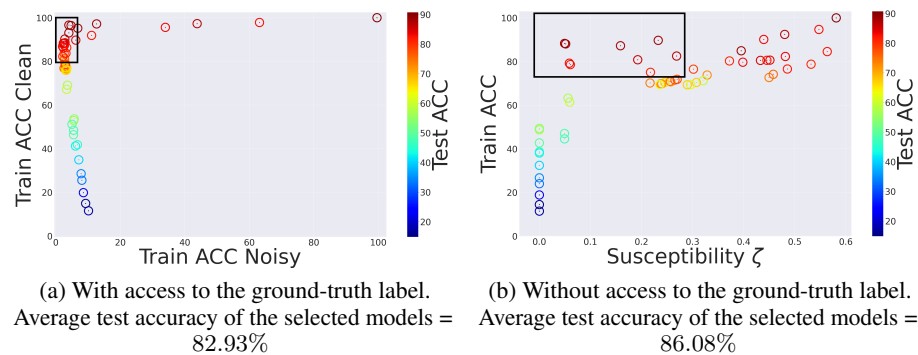

(a) With access to the ground-truth label. Average test accuracy of the selected models = $82.93\%$

(b) Without access to the ground-truth label. Average test accuracy of the selected models = $86.08\%$

Figure 34: For models trained on CIFAR-10 with $10\%$ label noise for 200 epochs, using susceptibility $\zeta$ and the overall training accuracy, the average test accuracy of the selected models is comparable with (even higher than) the case of having access to the ground-truth label.

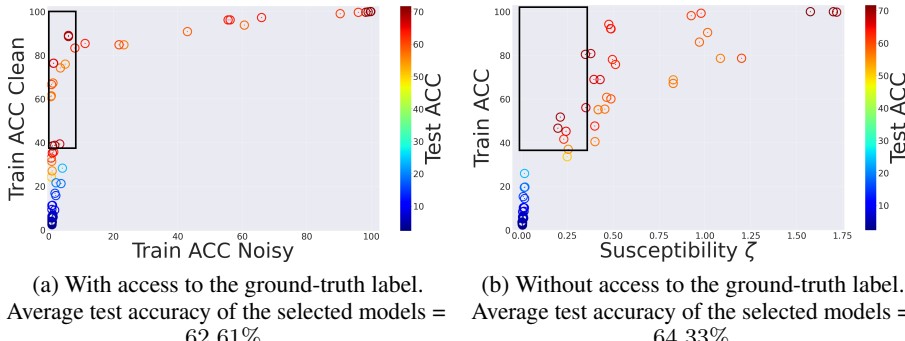

(a) With access to the ground-truth label. Average test accuracy of the selected models = 62.61%

(b) Without access to the ground-truth label. Average test accuracy of the selected models = 64.33%

Figure 35: For models trained on CIFAR-100 with $10\%$ label noise for 200 epochs, using susceptibility $\zeta$ and the overall training accuracy, the average test accuracy of the selected models is comparable with (even higher than) the case of having access to the ground-truth label.

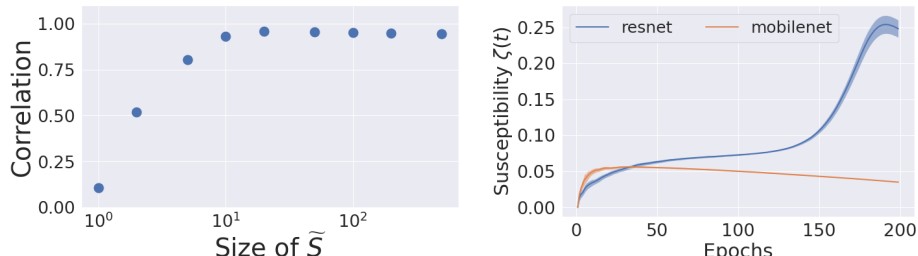

Figure 36: **Left**: Pearson correlation coefficient between the accuracy on the noisy subset of the training set and susceptibility $\zeta$ (Equation (2)) for different choices of dataset size for $\widetilde{S}$ for ResNet (He et al., 2016a), MobileNet (Howard et al., 2017), and $5-$layer cnn that are trained on CIFAR-100 dataset with $50\%$ label noise. We observe that unless the dataset is very small, the choice of the dataset size $\widetilde{S}$ does not affect the correlation value. Therefore, throughout our experiments, we choose the size 128 for this set, which is the batch size used for the regular training procedure as well. Note that this size is very small compared to the size of the training set itself, which is $50000$, hence the computational overhead to compute $\zeta$ is negligible compared to the original training process. **Right**: We can observe the variance of the susceptibility metric over 10 different random seeds. We can observe that as the variance is quite low, the metric is robust to the exact choice of the mini-batch and to the random labels that are assigned to the mini-batch.

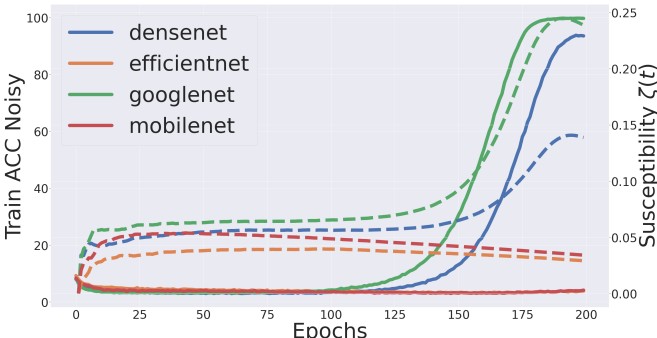

Figure 37: Accuracy on the noisy subset (solid lines) versus Susceptibility $\zeta(t)$ (dashed lines) for neural networks trained on CIFAR-10 with $50\%$ label noise. We observe a very strong match between the two, which suggests that susceptibility can be used to perform early stopping by selecting the checkpoint for each model with the least memorization. For example, for MobileNet and EfficientNet, $\zeta$ does not warn about memorization, hence one can select the end checkpoint. On the other hand, for DenseNet and GoogleNet, $\zeta$ suggests selecting those checkpoints that are before the sharp increases. This is also consistent with the signal given by the fit on the noisy subset, which requires ground-truth label access, unlike susceptibility $\zeta$ which does not require such access.

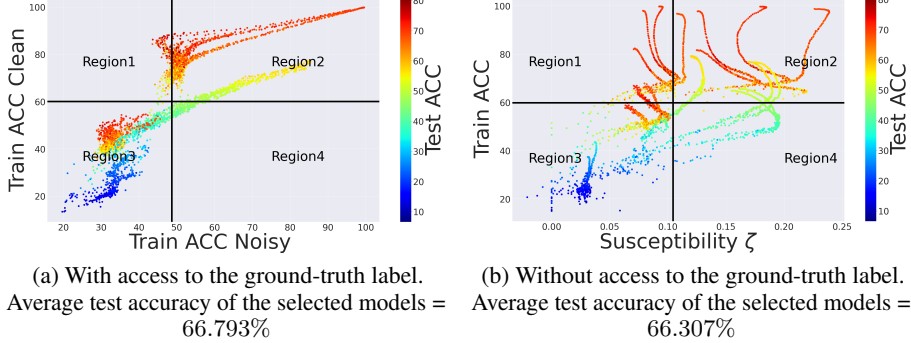

(a) With access to the ground-truth label. Average test accuracy of the selected models = $66.793\%$

(b) Without access to the ground-truth label. Average test accuracy of the selected models = $66.307\%$

Figure 38: For models trained for 200 epochs on CIFAR-10 with $50\%$ asymmetric label noise as proposed in (Xia et al., 2021), using susceptibility $\zeta$ and the overall training accuracy, the average test accuracy of the selected models is comparable with the case of having access to the ground-truth label.

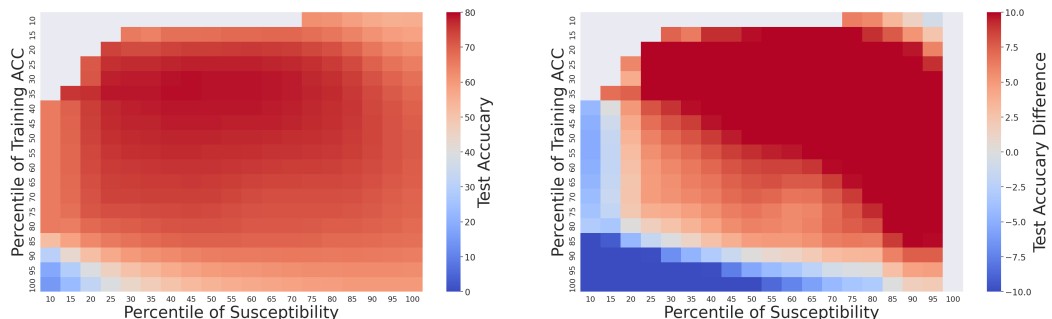

Figure 39: **Left:** Average test accuracy of models in Region 1 of Figure 5 for various thresholds used to find Region 1. In Figure 5 and throughout this paper, Region 1 has models with susceptibility $\zeta <$ t1 and training accuracy $>$ t2, where t1 and t2 are average $\zeta$ and training accuracy over the available models, respectively. Here, we study different values of these thresholds t1 and t2 and their effect on the average test accuracy of models of Region 1. We explore different percentiles of $\zeta$ and training accuracy over all models to be used to find these thresholds. The extreme would be to have 100th percentiles for both thresholds (low rightmost item of this table), which means models of Region 1 have $\zeta <$ maximum susceptibility and training accuracy $>$ minimum training accuracy. In this extreme case, all models are selected in Region 1. Overall, we observe that some other percentiles might give higher test accuracies than simply using the average values. These percentiles range however typically from 35 to 55 and are therefore not far from the mean, hence their benefit in increasing test accuracy appears small compared to the increased complexity to compute them or to rely on additional assumptions on the distribution of susceptibility and training accuracy. **Right:** The difference in the average test accuracies of models in Region 1 and models in Region 2 for various values of percentiles used to find different regions. A positive value implies that models in Region 1 have a higher average test accuracy. We can observe that except for very extreme values of the thresholds, which basically select all models as trainable, the average test accuracy of models in Region 1 is much higher than the average test accuracy of models in Region 2. Hence, our approach to select *resistant and trainable* models is robust to the choice of these thresholds.

## G   THEORETICAL PRELIMINARIES

In this section, we provide some technical tools that we use throughout our proofs. Recall from Section 2 that the first layer weights of the neural network are initialized as

$$\mathbf{w}_r(0) \sim \mathcal{N}\left(\mathbf{0}, \kappa^2\mathbf{I}\right), \quad \forall r \in [m], \tag{7}$$

where $0 < \kappa \leq 1$ is the magnitude of initialization and $\mathcal{N}$ denotes the normal distribution. And the second layer weights $a_r$s are independent random variables taking values uniformly in $\{-1, 1\}$ at initialization.

### G.1   PROPERTIES OF THE GRAM-MATRIX

**Properties**   Here, we recall a few useful properties of the Gram-matrix (Equation (3)).

1. As shown by (Du et al., 2018), $\mathbf{H}^\infty$ is positive definite and $\lambda_0 = \lambda_{\min}(\mathbf{H}^\infty) > 0$.
2. The matrix $\mathbf{H}^\infty$ has eigen decomposition $\mathbf{H}^\infty = \sum_{i=1}^n \lambda_i \mathbf{v}_i \mathbf{v}_i^T$, where the eigenvectors are orthonormal. Therefore, $\mathbf{v}_i^T \mathbf{v}_j = \delta_{i,j}$ for $i, j \in [n]$, the $n \times n$ identity matrix $\mathbf{I}$ is decomposed as $\sum_{i=1}^n \mathbf{v}_i \mathbf{v}_i^T$ and any $n$-dimensional (column-wise) vector $\mathbf{y}$ can be decomposed as $\mathbf{I}\,\mathbf{y} = \sum_{i=1}^n (\mathbf{v}_i^T \mathbf{y}) \mathbf{v}_i$.
3. (Recalled from (Du et al., 2018; Arora et al., 2019)) We have $\|\mathbf{H}^\infty\|_2 \leq \mathrm{tr}(\mathbf{H}^\infty) = \frac{n}{2} = \sum_{i=1}^n \lambda_i$, and

$$\eta = O(\frac{\lambda_0}{n^2}) = O\left(\frac{\lambda_{\min}(\mathbf{H}^\infty)}{\|\mathbf{H}^\infty\|_2^2}\right) \leq \frac{1}{\|\mathbf{H}^\infty\|_2}.$$

   Hence,

$$\|\mathbf{I} - \eta\mathbf{H}^\infty\|_2 \leq 1 - \eta\lambda_0.$$

### G.2   COROLLARIES ADAPTED FROM (DU ET AL., 2018; ARORA ET AL., 2019)

**Corollary 1.** *(Adapted Theorem 3.1 of (Arora et al., 2019) to our setting) For $m = \Omega\left(\frac{n^6}{\lambda_0^4 \kappa^2 \delta^3}\right)$ and $\eta = O\left(\frac{\lambda_0}{n^2}\right)$, for any $\delta \in (0, 1]$, with probability at least $1 - \delta$ over random initialization (7):*

$$\Phi\left(\boldsymbol{W}(0)\right) = O\left(\frac{n}{\delta}\right),$$

*and*

$$\begin{cases} \Phi\left(\boldsymbol{W}(t+1)\right) \leq (1 - \frac{\eta\lambda_0}{2})\Phi(\boldsymbol{W}(t)), & \text{if } 0 \leq t < k, \\ \widetilde{\Phi}\left(\boldsymbol{W}(t+1)\right) \leq (1 - \frac{\eta\lambda_0}{2})\widetilde{\Phi}(\boldsymbol{W}(t)), & \text{if } k \leq t < k + \tilde{k}. \end{cases}$$

Therefore, by replacing Equations (1) and (1), throughout the proof we can use:

$$\left\|\mathbf{f}_{\mathbf{W}(0)} - \mathbf{y}\right\|_2 = O\left(\sqrt{\frac{n}{\delta}}\right),$$

and

$$\begin{cases} \left\|\mathbf{f}_{\mathbf{W}(t+1)} - \mathbf{y}\right\|_2 & \leq \sqrt{1 - \frac{\eta\lambda_0}{2}} \left\|\mathbf{f}_{\mathbf{W}(t)} - \mathbf{y}\right\|_2 \\ & \leq \left(1 - \frac{\eta\lambda_0}{4}\right)\left\|\mathbf{f}_{\mathbf{W}(t)} - \mathbf{y}\right\|_2, & \text{if } 0 \leq t < k, \\ \left\|\mathbf{f}_{\mathbf{W}(t+1)} - \widetilde{\mathbf{y}}\right\|_2 & \leq \left(1 - \frac{\eta\lambda_0}{4}\right)\left\|\mathbf{f}_{\mathbf{W}(t)} - \widetilde{\mathbf{y}}\right\|_2, & \text{if } k \leq t < k + \tilde{k}, \end{cases}$$

where we use inequality $\sqrt{1 - \alpha} \leq 1 - \alpha/2$, which holds for $0 \leq \alpha \leq 1$.

**Corollary 2.** *(Adapted from Equation (25) of (Arora et al., 2019)) If the parameter vector is updated at step $t$ by one gradient descent step on $\frac{1}{2}\left\|\mathbf{f}_{\mathbf{W}(t)} - \mathbf{u}\right\|_2^2$ for some label vector $\mathbf{u}$, and for $t$ such that with probability at least $1 - \delta$:*

$$\|\mathbf{H}(t) - \mathbf{H}(0)\|_F = O\left(\frac{n^3}{\sqrt{m}\lambda_0 \kappa \delta^{3/2}}\right),$$

*then the output of the neural network is as follows*

$$\mathbf{f}_{\mathbf{W}(t+1)} - \mathbf{f}_{\mathbf{W}(t)} = -\eta \mathbf{H}^{\infty} \left( \mathbf{f}_{\mathbf{W}(t)} - \mathbf{u} \right) + \xi(t),$$

*where $\xi(\cdot)$ is considered to be a perturbation term that can be bounded with probability at least $1 - \delta$ over random initialization (7) by*

$$\|\xi(t)\|_2 = O\left( \frac{\eta n^3}{\sqrt{m}\lambda_0 \kappa \delta^{3/2}} \right) \|\mathbf{f}_{\mathbf{W}(t)} - \mathbf{u}\|_2. \tag{8}$$

Remark: In our setting, this corollary holds for $0 \leq t \leq k - 1$ with $\mathbf{u} = \mathbf{y}$, and for $k \leq t \leq k + \tilde{k} - 1$ with $\mathbf{u} = \widetilde{\mathbf{y}}$. We only need to show that for our setting for $t \geq k$, $\|\mathbf{H}(t) - \mathbf{H}(0)\|_F$ is bounded, which is done in Lemma 3.

**Corollary 3.** *(From Equation (27) of (Arora et al., 2019)) We have for $1 \leq t \leq k$*

$$\mathbf{f}_{\mathbf{W}(t)} - \mathbf{y} = (\mathbf{I} - \eta \mathbf{H}^{\infty})^t \left( \mathbf{f}_{\mathbf{W}(0)} - \mathbf{y} \right) + \sum_{s=0}^{t-1} (\mathbf{I} - \eta \mathbf{H}^{\infty})^s \xi\left( t - s - 1 \right),$$

*where $\|\xi(\cdot)\|_2$ is some perturbation term that can be bounded using Equation (8) with $\mathbf{u} = \mathbf{y}$.*

### G.3  ADDITIONAL LEMMAS

**Lemma 1.** *For the setting described in Section 2, we have*

$$\mathbf{f}_{\mathbf{W}(k)} - \widetilde{\mathbf{y}} = \sum_{i=1}^n \left[ (\mathbf{v}_i^T \mathbf{y}) - (1 - \eta \lambda_i)^k (\mathbf{v}_i^T \mathbf{y}) - (\mathbf{v}_i^T \widetilde{\mathbf{y}}) \right] \mathbf{v}_i + \chi(k), \tag{9}$$

*where $\chi(k)$ is some perturbation term that with probability at least $1 - \delta$ over the random initialization (7)*

$$\|\chi(k)\|_2 = O\left( \frac{n^{3/2}\kappa}{\sqrt{\delta}\lambda_0} + \frac{n^{9/2}}{\sqrt{m}\lambda_0^3 \kappa \delta^2} \right). \tag{10}$$

#### G.3.1  LEMMAS TO BOUND $\mathbf{H}(t)$ WITH $\mathbf{H}^{\infty}$

Because the two datasets $S$ and $\widetilde{S}$ have the same input samples, the Gram matrix defined in Equation (3) is the same for both of them. We now recall two lemmas from (Du et al., 2018; Arora et al., 2019) and provide a lemma extending them to bound $\mathbf{H}(t)$ with $\mathbf{H}^{\infty}$, where

$$\mathbf{H}_{ij}(t) = \frac{\mathbf{x}_i^T \mathbf{x}_j}{m} \sum_{r=1}^m \mathbb{I}_{r,i}(t)\mathbb{I}_{r,j}(t),$$

and $\mathbb{I}_{r,i}(t) = \mathbb{I}\{\mathbf{w}_r^T(t)\mathbf{x}_i \geq 0\}$.

**Lemma 2.** *(recalled from (Du et al., 2018; Arora et al., 2019)) For $\lambda_0 = \lambda_{min}(\boldsymbol{H}^{\infty}) > 0$, $m = \Omega\left( \frac{n^6}{\lambda_0^4 \kappa^2 \delta^3} \right)$, and $\eta = O\left( \frac{\lambda_0}{n^2} \right)$ with probability at least $1 - \delta$ over the random initialization (7), for all $0 \leq t \leq k$, we have:*

$$\|\mathbf{H}(t) - \mathbf{H}(0)\|_F = O\left( \frac{n^3}{\sqrt{m}\lambda_0 \kappa \delta^{3/2}} \right).$$

**Lemma 3.** *(Our extension) For $\lambda_0 = \lambda_{min}(\boldsymbol{H}^{\infty}) > 0$, $m = \Omega\left( \frac{n^6}{\lambda_0^4 \kappa^2 \delta^3} \right)$, and $\eta = O\left( \frac{\lambda_0}{n^2} \right)$ with probability at least $1 - \delta$ over the random initialization (7), for all $k + 1 \leq t \leq k + \tilde{k}$, we have:*

$$\|\mathbf{H}(t) - \mathbf{H}(0)\|_F = O\left( \frac{n^3}{\sqrt{m}\lambda_0 \kappa \delta^{3/2}} \right).$$

**Lemma 4.** *(recalled from (Du et al., 2018; Arora et al., 2019)) With probability at least $1 - \delta$ over the random initialization (7), we have:*

$$\|\mathbf{H}(0) - \mathbf{H}^\infty\|_F = O\left(\frac{n\sqrt{\log\frac{n}{\delta}}}{\sqrt{m}}\right).$$

Remark for Lemma 4: The indicator function $\mathbb{I}\{\mathbf{w}_r(t)^T\mathbf{x}_i \geq 0\}$ is invariant to the scale $\kappa$ of $\mathbf{w}_r$, hence $\mathbb{E}[\mathbf{H}_{ij}(0)] = \mathbf{H}_{ij}^\infty$, even though the expectation on the left hand side is taken with respect to $\mathbf{w} \sim \mathcal{N}(\mathbf{0}, \kappa^2\mathbf{I})$ and the expectation on the right hand side is taken with respect to $\mathbf{w} \sim \mathcal{N}(\mathbf{0}, \mathbf{I})$.

## H  PROOF OF LEMMA 3

*Proof.* We recall from the proof of Lemma C.2 of (Arora et al., 2019) that if with probability at least $1 - \delta$, $\|\mathbf{w}_r(t) - \mathbf{w}_r(0)\|_2 \leq R$, then with probability at least $1 - \delta$, we have $\|\mathbf{H}(t) - \mathbf{H}(0)\|_F \leq \frac{4n^2 R}{\sqrt{2\pi}\kappa\delta} + \frac{2n^2}{m}$. So, we first find an upper bound on $\|\mathbf{w}_r(t) - \mathbf{w}_r(0)\|_2$ for $t > k$ and replace its value $R$ in $\frac{4n^2 R}{\sqrt{2\pi}\kappa\delta} + \frac{2n^2}{m}$.

To find an upper bound on $\|\mathbf{w}_r(t) - \mathbf{w}_r(0)\|_2$ for $t > k$, we can follow a similar approach as in the proof of Lemma C.1 of (Arora et al., 2019):

$$\|\mathbf{w}_r(t) - \mathbf{w}_r(0)\|_2 \leq \sum_{\tau=0}^{t-1} \|\mathbf{w}_r(\tau+1) - \mathbf{w}_r(\tau)\|_2$$

$$= \sum_{\tau=0}^{k-1} \|\mathbf{w}_r(\tau+1) - \mathbf{w}_r(\tau)\|_2 + \sum_{\tau=k}^{t-1} \|\mathbf{w}_r(\tau+1) - \mathbf{w}_r(\tau)\|_2$$

$$\overset{(a)}{\leq} \sum_{\tau=0}^{k-1} \frac{\eta\sqrt{n}}{\sqrt{m}} \|\mathbf{f}_{\mathbf{W}(\tau)} - \mathbf{y}\|_2 + \sum_{\tau=k}^{t-1} \frac{\eta\sqrt{n}}{\sqrt{m}} \|\mathbf{f}_{\mathbf{W}(\tau)} - \widetilde{\mathbf{y}}\|_2$$

$$\overset{(b)}{\leq} \sum_{\tau=0}^{k-1} \frac{\eta\sqrt{n}}{\sqrt{m}} \left(1 - \frac{\eta\lambda_0}{4}\right)^\tau \|\mathbf{f}_{\mathbf{W}(0)} - \mathbf{y}\|_2$$

$$+ \sum_{\tau=k}^{t-1} \frac{\eta\sqrt{n}}{\sqrt{m}} \left(1 - \frac{\eta\lambda_0}{4}\right)^{\tau-k} \|\mathbf{f}_{\mathbf{W}(k)} - \widetilde{\mathbf{y}}\|_2$$

$$\overset{(c)}{\leq} O\left(\frac{\eta n}{\sqrt{m\delta}}\right) \sum_{\tau=0}^{k-1} \left(1 - \frac{\eta\lambda_0}{4}\right)^\tau$$

$$+ \sum_{\tau=k}^{t} \frac{\eta\sqrt{n}}{\sqrt{m}} \left(1 - \frac{\eta\lambda_0}{4}\right)^{\tau-k} \left[\|\mathbf{f}_{\mathbf{W}(k)} - \mathbf{y}\|_2 + \|\widetilde{\mathbf{y}} - \mathbf{y}\|_2\right]$$

$$\overset{(d)}{\leq} O\left(\frac{\eta n}{\sqrt{m\delta}}\right) \sum_{\tau=0}^{k-1} \left(1 - \frac{\eta\lambda_0}{4}\right)^\tau + O\left(\frac{\eta n}{\sqrt{m\delta}}\right) \sum_{\tau=0}^{t-k} \left(1 - \frac{\eta\lambda_0}{4}\right)^\tau$$

$$\leq O\left(\frac{\eta n}{\sqrt{m\delta}}\right) \sum_{\tau=0}^{\infty} \left(1 - \frac{\eta\lambda_0}{4}\right)^\tau \leq O\left(\frac{n}{\lambda_0\sqrt{m\delta}}\right), \tag{11}$$

where $(a)$ holds because for an update step on label vector $\mathbf{u}$ according to gradient descent, we have

$$\|\mathbf{w}_r(\tau+1) - \mathbf{w}_r(\tau)\| = \left\|\frac{\eta}{\sqrt{m}}a_r \sum_{i=1}^{n} \left(f_{\mathbf{W}(\tau)}(\mathbf{x}_i) - u_i\right)\mathbb{I}_{r,i}(k)\mathbf{x}_i\right\|$$

$$\leq \frac{\eta}{\sqrt{m}} \sum_{i=1}^{n} \left|f_{\mathbf{W}(\tau)}(\mathbf{x}_i) - u_i\right| \leq \frac{\eta\sqrt{n}}{\sqrt{m}} \|\mathbf{f}_{\mathbf{W}(\tau)} - \mathbf{u}\|,$$

inequalities $(b)$ and $(c)$ use Corollary 1. Inequality $(d)$ holds because we have $|y_i| \leq 1$ and $|\widetilde{y}_i| = 1$, so again by using Corollary 1

$$\left\|\mathbf{f}_{\mathbf{W}(k)} - \mathbf{y}\right\|_2 + \left\|\widetilde{\mathbf{y}} - \mathbf{y}\right\|_2 \leq \left(1 - \frac{\eta\lambda_0}{4}\right)^k O\left(\sqrt{\frac{n}{\delta}}\right) + O(\sqrt{n}) = O\left(\sqrt{\frac{n}{\delta}}\right). \tag{12}$$

Therefore, with probability at least $1 - \delta$ over random initialization (7) for $t > k$

$$\|\mathbf{H}(t) - \mathbf{H}(0)\|_F \leq \frac{4n^2 R}{\sqrt{2\pi}\kappa\delta} + \frac{2n^2}{m} = O\left(\frac{n^3}{\kappa\lambda_0\delta^{3/2}\sqrt{m}}\right),$$

where we replaced $R$ with the upper bound in Equation (11).

$\square$

## I   PROOF OF LEMMA 1

*Proof.* Note that throughout the proof, we refer to events with probability at least $1 - \delta$, as high probability events. Using the union bound, the probability of intersection of $\alpha$ high-probability events is an event with probability at least $1 - \alpha\delta$. Therefore, we can again refer to this event as a high probability event with probability at least $1 - \delta$, but re-scale $\delta$ in the event accordingly. Because $\delta$ only appears on bounds of the perturbation terms, and therefore in the form of $O(\delta^{-1})$, then re-scaling $\delta$ would not change the order of these perturbation terms. Hence, throughout the proof we do not put concerns on the exact probability of events, we only refer to them as high probability events, and eventually we know that the probability of our computations is at least $1 - \delta$ over the random initialization Equation (7).

Because Lemma 2 holds for $t = k - 1$, we use Corollary 2 with $t = k - 1$ and $\mathbf{u} = \mathbf{y}$:

$$\mathbf{f}_{\mathbf{W}(k)} - \mathbf{f}_{\mathbf{W}(k-1)} = -\eta\mathbf{H}^\infty\left(\mathbf{f}_{\mathbf{W}(k-1)} - \mathbf{y}\right) + \xi(k-1),$$

where with probability at least $1 - \delta$

$$\|\xi(k-1)\|_2 = O\left(\frac{\eta n^3}{\sqrt{m}\lambda_0\kappa\delta^{3/2}}\right)\left\|\mathbf{f}_{\mathbf{W}(k-1)} - \mathbf{y}\right\|_2,$$

because of Equation (8). We then compute

$$\mathbf{f}_{\mathbf{W}(k)} - \widetilde{\mathbf{y}} = \mathbf{f}_{\mathbf{W}(k-1)} - \widetilde{\mathbf{y}} - \eta\mathbf{H}^\infty\left(\mathbf{f}_{\mathbf{W}(k-1)} - \mathbf{y}\right) + \xi(k-1)$$

$$= \mathbf{f}_{\mathbf{W}(0)} - \widetilde{\mathbf{y}} - \eta\sum_{t=0}^{k-1}\mathbf{H}^\infty\left(\mathbf{f}_{\mathbf{W}(t)} - \mathbf{y}\right) + \sum_{t=0}^{k-1}\xi(t). \tag{13}$$

Let

$$\chi(k) = -\eta\sum_{t=0}^{k-1}\mathbf{H}^\infty\left(\mathbf{I} - \eta\mathbf{H}^\infty\right)^t \mathbf{f}_{\mathbf{W}(0)}$$

$$-\eta\sum_{t=1}^{k-1}\sum_{s=0}^{t-1}\mathbf{H}^\infty\left(\mathbf{I} - \eta\mathbf{H}^\infty\right)^s \xi\left(t - s - 1\right) + \sum_{t=0}^{k-1}\xi(t) + \mathbf{f}_{\mathbf{W}(0)}. \tag{14}$$

Then Equation (13) becomes:

$$\mathbf{f}_{\mathbf{W}(k)} - \widetilde{\mathbf{y}} = \eta\sum_{t=0}^{k-1}\mathbf{H}^\infty\left(\mathbf{I} - \eta\mathbf{H}^\infty\right)^t \mathbf{y} - \widetilde{\mathbf{y}} + \chi(k), \tag{15}$$

where we have replaced $\mathbf{f}_{\mathbf{W}(t)} - \mathbf{y}$ for $1 \leq t \leq k$ in Equation (13) using Corollary 3 by

$$\left(\mathbf{I} - \eta\mathbf{H}^\infty\right)^t \left(\mathbf{f}_{\mathbf{W}(0)} - \mathbf{y}\right) + \sum_{s=0}^{t-1}\left(\mathbf{I} - \eta\mathbf{H}^\infty\right)^s \xi(t - s - 1).$$

Next, we would like to find an upper bound for $\|\chi(k)\|$. As shown in Du et al. (2018); Arora et al. (2019), with probability at least $1 - \delta$ over random initialization (7), we have

$$\left\|\mathbf{f}_{\mathbf{W}(0)}\right\|_2^2 \leq \frac{n\kappa^2}{\delta}. \tag{16}$$

The first term in $\chi(k)$ (Equation (14)) is bounded because of Equation (16) from above with high probability as

$$\left\|\eta \sum_{t=0}^{k-1} \mathbf{H}^\infty \left(\mathbf{I} - \eta\mathbf{H}^\infty\right)^t \mathbf{f}_{\mathbf{W}(0)}\right\|_2 \leq \eta \sum_{t=0}^{k-1} \|\mathbf{H}^\infty\|_2 \|\mathbf{I} - \eta\mathbf{H}^\infty\|_2^t \left\|\mathbf{f}_{\mathbf{W}(0)}\right\|_2$$

$$\leq \eta \sum_{t=0}^{k-1} \frac{n}{2} (1 - \eta\lambda_0)^t O\left(\frac{\sqrt{n}\kappa}{\sqrt{\delta}}\right)$$

$$= O\left(\frac{n^{3/2}\kappa}{\sqrt{\delta}\lambda_0}\right),$$

where the second inequality uses Property 3.

The second term of $\chi(k)$ in Equation (14) can also be bounded with high probability by

$$\left\|\eta \sum_{t=1}^{k-1} \sum_{s=0}^{t-1} \mathbf{H}^\infty \left(\mathbf{I} - \eta\mathbf{H}^\infty\right)^s \xi(t-s-1)\right\|_2$$

$$\leq \eta \sum_{t=1}^{k-1} \sum_{s=0}^{t-1} \|\mathbf{H}^\infty\|_2 \|\mathbf{I} - \eta\mathbf{H}^\infty\|_2^s \|\xi(t-s-1)\|_2$$

$$\overset{(a)}{\leq} \eta \sum_{t=1}^{k-1} \sum_{s=0}^{t-1} \frac{n}{2} (1 - \eta\lambda_0)^s O\left(\frac{\eta n^3}{\sqrt{m}\lambda_0\kappa\delta^{3/2}}\right) \left\|\mathbf{f}_{\mathbf{W}(t-s-1)} - \mathbf{y}\right\|_2$$

$$\overset{(b)}{\leq} \eta \sum_{t=1}^{k-1} \sum_{s=0}^{t-1} \frac{n}{2} (1 - \eta\lambda_0)^s O\left(\frac{\eta n^3}{\sqrt{m}\lambda_0\kappa\delta^{3/2}}\right) \left(1 - \frac{\eta\lambda_0}{4}\right)^{t-s-1} O\left(\sqrt{\frac{n}{\delta}}\right)$$

$$= O\left(\frac{\eta^2 n^{9/2}}{\sqrt{m}\lambda_0\kappa\delta^2}\right) \sum_{t=1}^{k-1} \sum_{s=0}^{t-1} (1 - \eta\lambda_0)^s \left(1 - \frac{\eta\lambda_0}{4}\right)^{t-s-1}$$

$$= O\left(\frac{\eta^2 n^{9/2}}{\sqrt{m}\lambda_0\kappa\delta^2}\right) \sum_{t=1}^{k-1} \left(1 - \frac{\eta\lambda_0}{4}\right)^{t-1} \sum_{s=0}^{t-1} \left(\frac{1 - \eta\lambda_0}{1 - \eta\lambda_0/4}\right)^s$$

$$= O\left(\frac{\eta^2 n^{9/2}}{\sqrt{m}\lambda_0\kappa\delta^2}\right) \sum_{t=1}^{k-1} \left(1 - \frac{\eta\lambda_0}{4}\right)^{t-1} \frac{4 - \eta\lambda_0}{3\eta\lambda_0} \left[1 - \left(\frac{1 - \eta\lambda_0}{1 - \eta\lambda_0/4}\right)^t\right]$$

$$= O\left(\frac{\eta^2 n^{9/2}}{\sqrt{m}\lambda_0\kappa\delta^2}\right) \frac{(4 - \eta\lambda_0)}{(3\eta\lambda_0)(1 - \eta\lambda_0/4)} \sum_{t=1}^{k-1} \left[\left(1 - \frac{\eta\lambda_0}{4}\right)^t - (1 - \eta\lambda_0)^t\right]$$

$$= O\left(\frac{\eta^2 n^{9/2}}{\sqrt{m}\lambda_0\kappa\delta^2}\right) \frac{4}{3\eta\lambda_0} \left[\frac{1 - (1 - \eta\lambda_0/4)^k}{\eta\lambda_0/4} - \frac{1 - (1 - \eta\lambda_0)^k}{\eta\lambda_0}\right]$$

$$\leq O\left(\frac{n^{9/2}}{\sqrt{m}\lambda_0^3\kappa\delta^2}\right),$$

where $(a)$ uses Property 3 and Equation (8), $(b)$ uses Corollary 1, and the rest of the computations use algebraic tricks.

The third term of $\chi(k)$ in Equation (14) can be bounded with high probability using Equation (8) by:

$$\left\| \sum_{t=0}^{k-1} \xi(t) \right\|_2 \le \sum_{t=0}^{k-1} \|\xi(t)\|_2 \le \sum_{t=0}^{k-1} O\left( \frac{\eta n^3}{\sqrt{m}\lambda_0 \kappa \delta^{3/2}} \right) \|\mathbf{f}_{\mathbf{W}(t)} - \mathbf{y}\|_2$$

$$\le O\left( \frac{\eta n^3}{\sqrt{m}\lambda_0 \kappa \delta^{3/2}} \right) O\left( \sqrt{\frac{n}{\delta}} \right) \sum_{t=0}^{k-1} \left( 1 - \frac{\eta \lambda_0}{4} \right)^t$$

$$\le O\left( \frac{n^{7/2}}{\sqrt{m}\lambda_0^2 \kappa \delta^2} \right),$$

where we use Corollary 1.

Summing up, and using Equation (16) to bound the last term of Equation (14), we showed that with high probability:

$$\|\chi(k)\|_2 \le O\left( \frac{n^{3/2}\kappa}{\sqrt{\delta}\lambda_0} + \frac{n^{9/2}}{\sqrt{m}\lambda_0^3 \kappa \delta^2} + \frac{n^{7/2}}{\sqrt{m}\lambda_0^2 \kappa \delta^2} + \frac{\sqrt{n}\kappa}{\sqrt{\delta}} \right)$$

$$= O\left( \frac{n^{3/2}\kappa}{\sqrt{\delta}\lambda_0} + \frac{n^{9/2}}{\sqrt{m}\lambda_0^3 \kappa \delta^2} \right). \tag{17}$$

We now reformulate Equation (15) in terms of the eigenvectors and eigenvalues of the Gram matrix (Equation (3)) as follows, by repeatedly using Property 2

$$\mathbf{f}_{\mathbf{W}(k)} - \widetilde{\mathbf{y}} = \eta \sum_{t=0}^{k-1} \mathbf{H}^\infty (\mathbf{I} - \eta \mathbf{H}^\infty)^t \mathbf{y} - \widetilde{\mathbf{y}} + \chi(k)$$

$$= \eta \sum_{t=0}^{k-1} \sum_{i=1}^{n} \lambda_i \mathbf{v}_i \mathbf{v}_i^T \sum_{j=1}^{n} (1 - \eta \lambda_j)^t \mathbf{v}_j \mathbf{v}_j^T \sum_{z=1}^{n} (\mathbf{v}_z^T \mathbf{y}) \mathbf{v}_z - \sum_{i=1}^{n} (\mathbf{v}_i^T \widetilde{\mathbf{y}}) \mathbf{v}_i + \chi(k)$$

$$= \eta \sum_{t=0}^{k-1} \sum_{i=1}^{n} \sum_{j=1}^{n} \sum_{z=1}^{n} \lambda_i (1 - \eta \lambda_j)^t \mathbf{v}_i (\mathbf{v}_i^T \mathbf{v}_j)(\mathbf{v}_j^T \mathbf{v}_z)(\mathbf{v}_z^T \mathbf{y}) - \sum_{i=1}^{n} (\mathbf{v}_i^T \widetilde{\mathbf{y}}) \mathbf{v}_i + \chi(k)$$

$$\overset{(a)}{=} \eta \sum_{t=0}^{k-1} \sum_{i=1}^{n} \sum_{j=1}^{n} \sum_{z=1}^{n} \lambda_i (1 - \eta \lambda_j)^t \mathbf{v}_i \delta_{i,j} \delta_{j,z} (\mathbf{v}_z^T \mathbf{y}) - \sum_{i=1}^{n} (\mathbf{v}_i^T \widetilde{\mathbf{y}}) \mathbf{v}_i + \chi(k)$$

$$= \eta \sum_{t=0}^{k-1} \sum_{i=1}^{n} \lambda_i (1 - \eta \lambda_i)^t \mathbf{v}_i (\mathbf{v}_i^T \mathbf{y}) - \sum_{i=1}^{n} (\mathbf{v}_i^T \widetilde{\mathbf{y}}) \mathbf{v}_i + \chi(k)$$

$$= \eta \sum_{i=1}^{n} \lambda_i \sum_{t=0}^{k-1} (1 - \eta \lambda_i)^t (\mathbf{v}_i^T \mathbf{y}) \mathbf{v}_i - \sum_{i=1}^{n} (\mathbf{v}_i^T \widetilde{\mathbf{y}}) \mathbf{v}_i + \chi(k)$$

$$= \sum_{i=1}^{n} \left[ 1 - (1 - \eta \lambda_i)^k \right] (\mathbf{v}_i^T \mathbf{y}) \mathbf{v}_i - \sum_{i=1}^{n} (\mathbf{v}_i^T \widetilde{\mathbf{y}}) \mathbf{v}_i + \chi(k)$$

$$= \sum_{i=1}^{n} \left[ (\mathbf{v}_i^T \mathbf{y}) - (1 - \eta \lambda_i)^k (\mathbf{v}_i^T \mathbf{y}) - (\mathbf{v}_i^T \widetilde{\mathbf{y}}) \right] \mathbf{v}_i + \chi(k), \tag{18}$$

where in $(a)$ with some abuse of notation $\delta_{i,j}$ and $\delta_{j,z}$ refer to the Kronecker delta function. This concludes the proof. $\qquad \square$

## J   PROOF OF THEOREM 1

*Proof.* We start similarly to the proof of Lemma 1. Because for $t = k + \tilde{k} - 1$ Lemma 3 holds, using Corollary 2 for $t = k + \tilde{k} - 1$ and $\mathbf{u} = \widetilde{\mathbf{y}}$, we have:

$$\mathbf{f}_{\mathbf{W}(k+\tilde{k})} - \mathbf{f}_{\mathbf{W}(k+\tilde{k}-1)} = -\eta \mathbf{H}^\infty \left( \mathbf{f}_{\mathbf{W}(k+\tilde{k}-1)} - \widetilde{\mathbf{y}} \right) + \xi(k + \tilde{k} - 1),$$

where from Equation (8) with high probability

$$\left\| \xi(k + \tilde{k} - 1) \right\|_2 = O\left( \frac{\eta n^3}{\sqrt{m}\lambda_0 \kappa \delta^{3/2}} \right) \left\| \mathbf{f}_{\mathbf{W}(k+\tilde{k}-1)} - \widetilde{\mathbf{y}} \right\|_2.$$

Therefore, recursively we can write:

$$
\begin{aligned}
\mathbf{f}_{\mathbf{W}(k+\tilde{k})} - \widetilde{\mathbf{y}} &= \mathbf{f}_{\mathbf{W}(k+\tilde{k}-1)} - \widetilde{\mathbf{y}} - \eta \mathbf{H}^\infty \left( \mathbf{f}_{\mathbf{W}(k+\tilde{k}-1)} - \widetilde{\mathbf{y}} \right) + \xi(k + \tilde{k} - 1) \\
&= (\mathbf{I} - \eta \mathbf{H}^\infty) \left( \mathbf{f}_{\mathbf{W}(k+\tilde{k}-1)} - \widetilde{\mathbf{y}} \right) + \xi(k + \tilde{k} - 1) \\
&= (\mathbf{I} - \eta \mathbf{H}^\infty)^{\tilde{k}} \left( \mathbf{f}_{\mathbf{W}(k)} - \widetilde{\mathbf{y}} \right) + \sum_{t=0}^{\tilde{k}-1} (\mathbf{I} - \eta \mathbf{H}^\infty)^t \xi(k + \tilde{k} - 1 - t) \\
&= (\mathbf{I} - \eta \mathbf{H}^\infty)^{\tilde{k}} \left[ \mathbf{y} - \widetilde{\mathbf{y}} - \sum_{i=1}^n (1 - \eta\lambda_i)^k \left( \mathbf{v}_i^T \mathbf{y} \right) \mathbf{v}_i + \chi(k) \right] \\
&\quad + \sum_{t=0}^{\tilde{k}-1} (\mathbf{I} - \eta \mathbf{H}^\infty)^t \xi(k + \tilde{k} - 1 - t) \\
&= (\mathbf{I} - \eta \mathbf{H}^\infty)^{\tilde{k}} \left[ \mathbf{y} - \widetilde{\mathbf{y}} - \sum_{i=1}^n (1 - \eta\lambda_i)^k \left( \mathbf{v}_i^T \mathbf{y} \right) \mathbf{v}_i \right] \\
&\quad + (\mathbf{I} - \eta \mathbf{H}^\infty)^{\tilde{k}} \chi(k) + \sum_{t=0}^{\tilde{k}-1} (\mathbf{I} - \eta \mathbf{H}^\infty)^t \xi\left( k + \tilde{k} - 1 - t \right),
\end{aligned}
\tag{19}
$$

where the 4th operation follows from Lemma 1. Now, we find bounds for the perturbation terms. Let

$$\kappa = O(\frac{\epsilon\sqrt{\delta}\lambda_0}{n^{3/2}}), \qquad\qquad m = \Omega(\frac{n^9}{\lambda_0^6 \epsilon^2 \kappa^2 \delta^4}). \tag{20}$$

Then, using Lemma 1, the first perturbation term of Equation (19) is upper bounded as

$$
\begin{aligned}
\left\| (\mathbf{I} - \eta \mathbf{H}^\infty)^{\tilde{k}} \chi(k) \right\|_2 &\leq (1 - \eta\lambda_0)^{\tilde{k}} O\left( \frac{n^{3/2}\kappa}{\sqrt{\delta}\lambda_0} + \frac{n^{9/2}}{\sqrt{m}\lambda_0^3 \kappa \delta^2} \right) \\
&= O\left( (1 - \eta\lambda_0)^{\tilde{k}} \epsilon \right) \in O(\epsilon),
\end{aligned}
$$

where the last line comes from inserting our choices of $\kappa$ and $m$ from Equation (20).
The last term of Equation (19) can be upper bounded with high probability as

$$\left\| \sum_{t=0}^{\tilde{k}-1} (\mathbf{I} - \eta \mathbf{H}^\infty)^t \xi(k + \tilde{k} - 1 - t) \right\| \le \sum_{t=0}^{\tilde{k}-1} \|\mathbf{I} - \eta \mathbf{H}^\infty\|_2^t \left\| \xi(k + \tilde{k} - 1 - t) \right\|_2$$

$$\overset{(a)}{\le} \sum_{t=0}^{\tilde{k}-1} (1 - \eta \lambda_0)^t \, O\left( \frac{\eta n^3}{\sqrt{m} \lambda_0 \kappa \delta^{3/2}} \right) \left\| \mathbf{f}_{\mathbf{W}(k+\tilde{k}-1-t)} - \widetilde{\mathbf{y}} \right\|_2$$

$$\overset{(b)}{\le} O\left( \frac{\eta n^3}{\sqrt{m} \lambda_0 \kappa \delta^{3/2}} \right) \sum_{t=0}^{\tilde{k}-1} (1 - \eta \lambda_0)^t \left( 1 - \frac{\eta \lambda_0}{4} \right)^{\tilde{k}-1-t} \left\| \mathbf{f}_{\mathbf{W}(k)} - \widetilde{\mathbf{y}} \right\|_2$$

$$\le O\left( \frac{\eta n^3}{\sqrt{m} \lambda_0 \kappa \delta^{3/2}} \right) \sum_{t=0}^{\tilde{k}-1} (1 - \eta \lambda_0)^t \left( 1 - \frac{\eta \lambda_0}{4} \right)^{\tilde{k}-1-t} \left[ \left\| \mathbf{f}_{\mathbf{W}(k)} - \mathbf{y} \right\|_2 + \left\| \widetilde{\mathbf{y}} - \mathbf{y} \right\|_2 \right]$$

$$\overset{(c)}{\le} O\left( \frac{\eta n^3}{\sqrt{m} \lambda_0 \kappa \delta^{3/2}} \right) \sum_{t=0}^{\tilde{k}-1} (1 - \eta \lambda_0)^t \left( 1 - \frac{\eta \lambda_0}{4} \right)^{\tilde{k}-1-t} \left[ O\left( \sqrt{\frac{n}{\delta}} \right) \right]$$

$$\le O\left( \frac{\eta n^{7/2}}{\sqrt{m} \lambda_0 \kappa \delta^2} \right) \left( 1 - \frac{\eta \lambda_0}{4} \right)^{\tilde{k}-1} \sum_{t=0}^{\tilde{k}-1} \left( \frac{1 - \eta \lambda_0}{1 - \eta \lambda_0/4} \right)^t$$

$$\le \frac{4}{3 \eta \lambda_0} O\left( \frac{\eta n^{7/2}}{\sqrt{m} \lambda_0 \kappa \delta^2} \right) = O\left( \frac{n^{7/2}}{\sqrt{m} \lambda_0^2 \kappa \delta^2} \right) \overset{(d)}{=} O\left( \frac{\lambda_0 \epsilon}{n} \right),$$

where $(a)$ uses Property 3 and Equation (8), $(b)$ uses Corollary 1 and $(c)$ uses Equation (12). Finally $(d)$ follows from inserting our choice of $m$ from Equation (20). Because we have $\sum_{i=1}^n \lambda_i = n/2$ from Property 3, and $\lambda_0 = \min\{\lambda_i\}_i^n$, then $\lambda_0 \le 1/2$. Both perturbation terms in Equation (19) are therefore at most in the order of $\epsilon$ with our choices of $m$ and $\kappa$ from Equation (20).

Using Property 2, the squared norm of the first term in Equation (19) is

$$\left\| \sum_{i=1}^n (1 - \eta \lambda_i)^{\tilde{k}} \left[ (\mathbf{v}_i^T \mathbf{y}) - (1 - \eta \lambda_i)^k (\mathbf{v}_i^T \mathbf{y}) - (\mathbf{v}_i^T \widetilde{\mathbf{y}}) \right] \mathbf{v}_i \right\|_2^2$$

$$= \sum_{i=1}^n \sum_{j=1}^n (1 - \eta \lambda_i)^{\tilde{k}} \left[ (\mathbf{v}_i^T \mathbf{y}) - (1 - \eta \lambda_i)^k (\mathbf{v}_j^T \mathbf{y}) - (\mathbf{v}_j^T \widetilde{\mathbf{y}}) \right]$$

$$(1 - \eta \lambda_j)^{\tilde{k}} \left[ (\mathbf{v}_j^T \mathbf{y}) - (1 - \eta \lambda_j)^k (\mathbf{v}_j^T \mathbf{y}) - (\mathbf{v}_j^T \widetilde{\mathbf{y}}) \right] \mathbf{v}_i^T \mathbf{v}_j$$

$$= \sum_{i=1}^n \left[ (\mathbf{v}_i^T \mathbf{y}) - (1 - \eta \lambda_i)^k (\mathbf{v}_j^T \mathbf{y}) - (\mathbf{v}_j^T \widetilde{\mathbf{y}}) \right]^2 (1 - \eta \lambda_i)^{2\tilde{k}} \tag{21}$$

The norm of Equation (19) is therefore, for our choice of $\kappa$ and $m$ given in Equation (20), with probability at least $1 - \delta$

$$\left\| \mathbf{f}_{\mathbf{W}(k+\tilde{k})} - \widetilde{\mathbf{y}} \right\|_2 = \sqrt{\sum_{i=1}^n \left[ \mathbf{v}_i^T \mathbf{y} - \mathbf{v}_i^T \widetilde{\mathbf{y}} - (1 - \eta \lambda_i)^k \mathbf{v}_i^T \mathbf{y} \right]^2 (1 - \eta \lambda_i)^{2\tilde{k}}} \pm \epsilon,$$

which concludes the proof. $\qquad\square$

## K  PROOF AND NUMERICAL EVALUATIONS OF THEOREM 2

*Proof.* Because $\widetilde{y}_j \sim U(\{-1, 1\})$ and $\widetilde{y}_j \perp\!\!\!\perp \widetilde{y}_i$ for $i \neq j$, and $\|\mathbf{v}_i\|_2 = 1$, for $i \in [n]$, we have:

$$\mathbb{E}\left[ \tilde{p}_i^2 \right] = \mathbb{E}\left[ \sum_{j=1}^n \mathbf{v}_{i,j}^2 \widetilde{y}_j^2 + \sum_{j=1}^n \sum_{\substack{k=1 \\ k \neq j}}^n \mathbf{v}_{i,j} \widetilde{y}_j \mathbf{v}_{i,k} \widetilde{y}_k \right] = \sum_{j=1}^n \mathbf{v}_{i,j}^2 + \sum_{j=1}^n \sum_{\substack{k=1 \\ k \neq j}}^n \mathbf{v}_{i,j} \mathbf{v}_{i,k} \mathbb{E}\left[ \widetilde{y}_j \right] \mathbb{E}\left[ \widetilde{y}_k \right] = 1 + 0 = 1.$$

Recall from Equation (4) that

$$\widetilde{\Phi}(k + \tilde{k}) = \frac{1}{2} \sum_{i=1}^{n} \left[ p_i - \tilde{p}_i - p_i \left(1 - \eta\lambda_i\right)^k \right]^2 \left(1 - \eta\lambda_i\right)^{2\tilde{k}} .$$

This expression is a random variable that depends on random vectors $\widetilde{\mathbf{p}}$ and $\mathbf{p}$, which are functions of the random label vectors $\widetilde{\mathbf{y}}$ and $\mathbf{y}$, respectively. We now compute the expectation of the above objective function with respect to $\widetilde{\mathbf{p}}$ and $\mathbf{p}$:

$$\mathbb{E}_{\widetilde{\mathbf{p}},\mathbf{p}} \left[ \widetilde{\Phi}(k + \tilde{k}) \right] = \frac{1}{2} \sum_{i=1}^{n} \mathbb{E}\left[p_i^2\right] \left[1 - (1 - \eta\lambda_i)^k\right]^2 (1 - \eta\lambda_i)^{2\tilde{k}}$$

$$+ \frac{1}{2} \sum_{i=1}^{n} \mathbb{E}\left[\tilde{p}_i^2\right] (1 - \eta\lambda_i)^{2\tilde{k}} - \sum_{i=1}^{n} \mathbb{E}\left[p_i \tilde{p}_i\right] \left[1 - (1 - \eta\lambda_i)^k\right] (1 - \eta\lambda_i)^{2\tilde{k}}$$

$$= \frac{1}{2} \sum_{i=1}^{n} \mathbb{E}\left[p_i^2\right] \left[1 - (1 - \eta\lambda_i)^k\right]^2 (1 - \eta\lambda_i)^{2\tilde{k}} + \frac{1}{2} \sum_{i=1}^{n} (1 - \eta\lambda_i)^{2\tilde{k}}$$

$$- \sum_{i=1}^{n} \left[ \sum_{j=1}^{n} \sum_{k=1}^{n} \mathbf{v}_{i,j} \mathbf{v}_{i,k} \mathbb{E}\left[\tilde{y}_j y_k\right] \right] \left[1 - (1 - \eta\lambda_i)^k\right] (1 - \eta\lambda_i)^{2\tilde{k}}$$

$$= \frac{1}{2}\mu + \frac{1}{2} \sum_{i=1}^{n} (1 - \eta\lambda_i)^{2\tilde{k}}, \tag{22}$$

where follows with $\mu$ given by Equation (5), and because $\widetilde{y}_j \perp\!\!\!\perp y_k$ for all $j, k \in [n]$.

Because of Chebyshev inequality and Equation (22), with probability at least $1 - \delta$, we have:

$$\left| \widetilde{\Phi}(k + \tilde{k}) - \frac{1}{2} \sum_{i=1}^{n} (1 - \eta\lambda_i)^{2\tilde{k}} - \frac{\mu}{2} \right| \leq \sqrt{\frac{\Sigma}{\delta}}, \tag{23}$$

where

$$\Sigma = \mathrm{Var}_{\widetilde{\mathbf{p}},\mathbf{p}}\left[ \widetilde{\Phi}(k + \tilde{k}) \right], \tag{24}$$

which concludes the proof.

$\square$

**Numerical Evaluations** We now empirically evaluate the lower and upper bounds in Equation (23) for networks trained on label vector $\mathbf{y}$ with varying label noise levels (LNL). To do so, we discard the middle term of the left hand side of Equation (23), as it does not depend on $\mathbf{y}$. We then study the rest in Figures 40, 41, 42 and 43 for different datasets and values of $\eta$ and $k$. We observe consistently that both the lower and the upper bounds are a decreasing function of the label noise level (LNL) in the label vector $\mathbf{y}$.

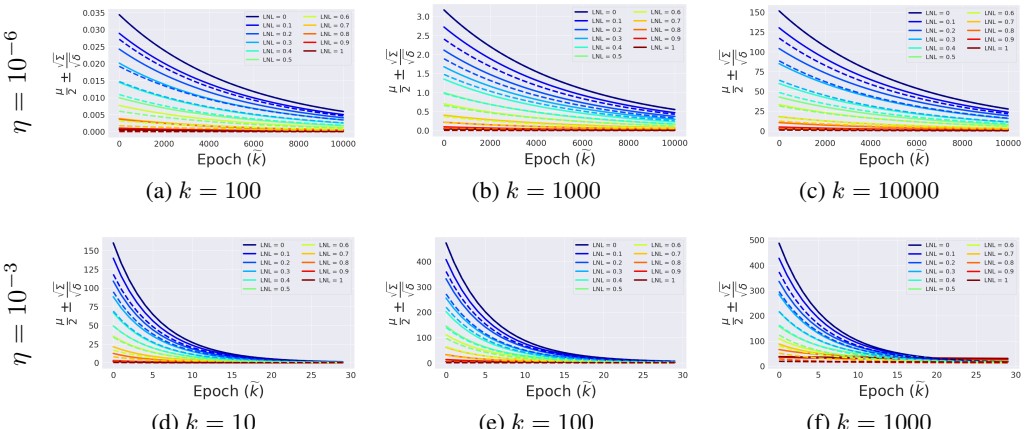

Figure 40: The lower (dashed lines) and upper (solid lines) bound terms of Theorem 2 that depend on the label noise level (LNL) are depicted as a function of the number of epochs $\widetilde{k}$, for different hyper-parameter values (the learning rate $\eta$ and $k$) with $\delta = 0.05$. The eigenvector projections $p_i$ that appear in $\mu$ and $\Sigma$ are computed from the Gram-matrix of 1000 samples from the MNIST dataset. The resulting values are obtained from an average over 10 random draws of the label vector $\mathbf{y}$. We observe that both the lower and the upper bounds of Equation (23) are decreasing functions of LNL and of $\widetilde{k}$.

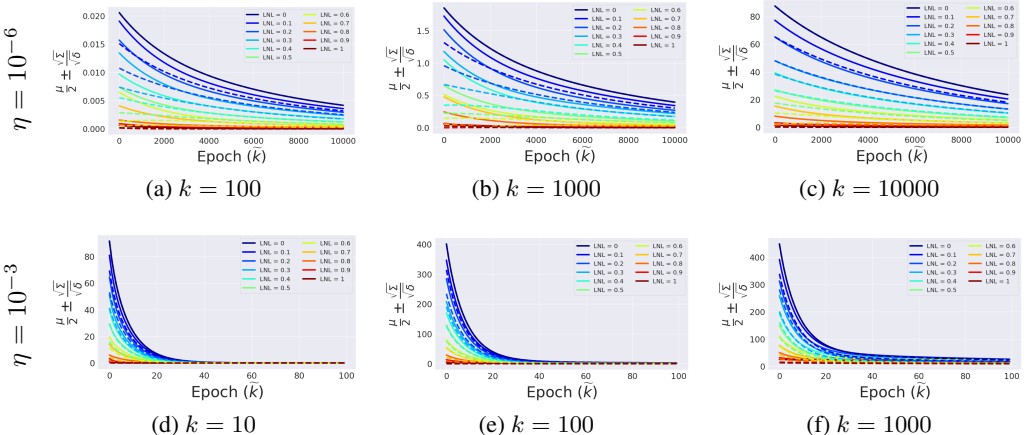

Figure 41: The lower (dashed lines) and upper (solid lines) bound terms of Theorem 2 that depend on the label noise level (LNL) are depicted as a function of the number of epochs $\widetilde{k}$, for different hyper-parameter values (the learning rate $\eta$ and $k$) with $\delta = 0.05$. The eigenvector projections $p_i$ that appear in $\mu$ and $\Sigma$ are computed from the Gram-matrix of 1000 samples from the Fashion-MNIST dataset. The resulting values are obtained from an average over 10 random draws of the label vector $\mathbf{y}$. We observe that both the lower and the upper bounds of Equation (23) are decreasing functions of LNL and of $\widetilde{k}$.

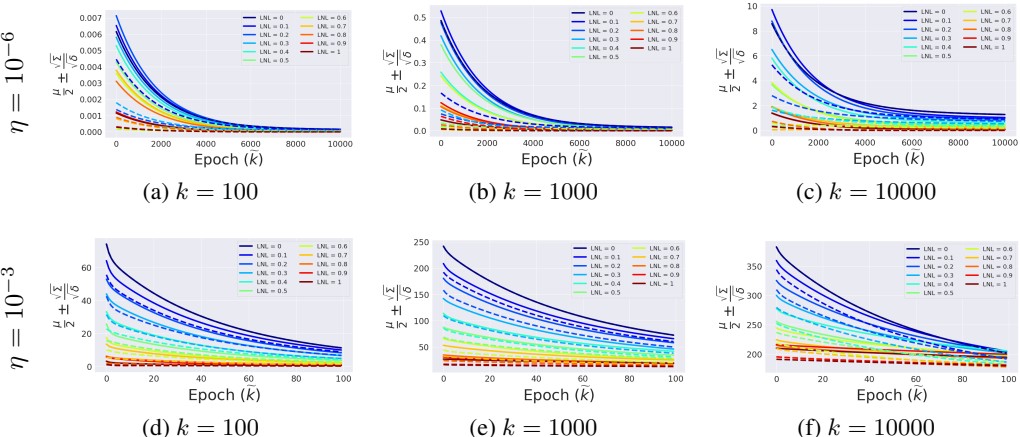

Figure 42: The lower (dashed lines) and upper (solid lines) bound terms of Theorem 2 that depend on the label noise level (LNL) are depicted as a function of the number of epochs $\widetilde{k}$, for different hyper-parameter values (the learning rate $\eta$ and $k$) with $\delta = 0.05$. The eigenvector projections $p_i$ that appear in $\mu$ and $\Sigma$ are computed from the Gram-matrix of 1000 samples from the CIFAR-10 dataset. The resulting values are obtained from an average over 10 random draws of the label vector $\mathbf{y}$. We observe that both the lower and the upper bounds of Equation (23) are decreasing functions of LNL and of $\widetilde{k}$.

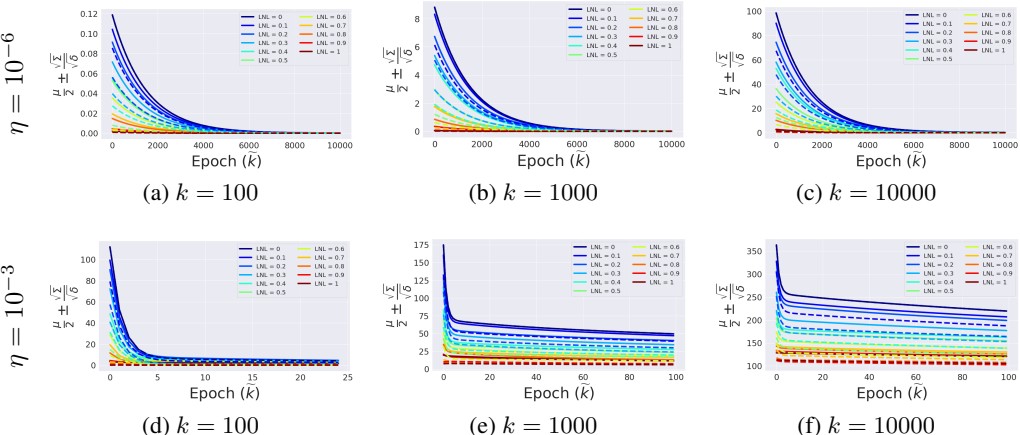

Figure 43: The lower (dashed lines) and upper (solid lines) bound terms of Theorem 2 that depend on the label noise level (LNL) are depicted as a function of the number of epochs $\widetilde{k}$, for different hyper-parameter values (the learning rate $\eta$ and $k$) with $\delta = 0.05$. The eigenvector projections $p_i$ that appear in $\mu$ and $\Sigma$ are computed from the Gram-matrix of 1000 samples from the SVHN dataset. The resulting values are obtained from an average over 10 random draws of the label vector $\mathbf{y}$. We observe that both the lower and the upper bounds of Equation (23) are decreasing functions of LNL and of $\widetilde{k}$.

