# OpenReview forum: "Leveraging Unlabeled Data to Track Memorization"
_ICLR.cc/2023/Conference — ICLR 2023 poster_

### Official Review · Reviewer_QKAt · 2022-10-24

**Confidence:** 3
**Correctness:** 3
**Technical Novelty And Significance:** 3
**Empirical Novelty And Significance:** 2
**Recommendation:** 6

**Clarity, Quality, Novelty And Reproducibility:**

- This paper is well-motivated and organized.
- The proposed susceptibility is novel. I have not seen it in other papers.
- The paper provides the code.

**Strength And Weaknesses:**

**Strengeth:**

- Using random labeled dataset to track memorization is new and interesting. The efficiency of susceptibility is supported by extensive experiments on multiple datasets including synthetic label noise and real-world label noise.

- The proposed metric is supported by theoretical analysis.

**Weakness:**

- It would be better to perform some experiments based on instance-dependent label noise [R1].

- This paper proves that a model with higher acc and lower susceptibility has better performance. But in pratice, it is hard to decide when should I stop training based on susceptibility and training acc. It seems one needs to set some specific thresholds to guide decision. Why noisy validation is not enough to guide decision since [R2] shows that the acc of noisy set is also robust.


[R1] Learning with bounded instance-and label-dependent label noise.

[R2] Robustness of accuracy metric and its inspirations in learning with noisy labels.


**Summary Of The Paper:**

It has been an convention that uses noisy validation seperated from training set to determine the model performance trained on dataset with label noise. The authors claim that noisy validation is not efficient and thus proposes a new evaluation protocal based on the susceptibility which meaures the difficulty for model to fit on a random labeled set.  Combining "susceptibility" with training accuracy can select good models that performs well on clean test set.

**Summary Of The Review:**

The proposed metric is intersting. The results make sense but I find it hard to use the proposed metric to guide training in learning with noisy labels.

---

> ### Author Response · Authors · 2022-11-15
> **Response to Reviewer QKAt**
>
> Thank you for your review. We are glad that you are satisfied with the motivation, organization, and novelty of our work and the support we provide for the proposed metric with extensive empirical investigations and theoretical results. Below we address the comments and questions you’ve raised, and we’d be very open to further discussion. We would appreciate it if you could increase your score if you are satisfied with our responses.
>
> **On instance-dependent label noise:**
>
> In our submission, we have reported results from two datasets with real-world label noise, namely Clothing-1M and Animal-10N, whose label noise is instance-dependent as pointed out in [2]. The results are presented in Appendix E.1 and show a consistently good performance for model selection. In addition, throughout this rebuttal phase,  we have included additional experiments on the CIFAR-10N dataset proposed in [1], which includes real-world noisy labels which are instance-dependent. In Figures 27-29 (c) in Appendix E.1 of the revised paper, we observe the success of our model selection approach in this dataset as well. Therefore, our approach extends to instance-dependent label noise types as well.
>
> **On finding thresholds in practice:**
>
> The thresholds are not manually set, e.g., by using hyper-parameter tuning, but are simply computed on the fly as the average values of the training accuracy and susceptibility over the available trained model. Therefore, setting these thresholds does not require any additional tuning, and does not bring any additional computational overhead.
> Please also refer to the common response to all reviewers for more details on these thresholds.
>
> **On comparison to a noisy validation set:**
>
> In the experiments in Table 1 and Figure 10, we observe that the noisy validation accuracy may produce low correlation values to the clean test accuracy, depending on the label noise level of the dataset and the size of the noisy validation set. In contrast, our approach consistently produces high values of correlations in all settings. In addition to robustness against label noise level and dataset size, our approach brings advantages in offering a gauge on memorization, without needing extra data, while maintaining a very low computational cost. Please refer to the common response to all reviewers for more details on this comparison.
>
> **On using Susceptibility to guide training:**
>
> Susceptibility is proposed to guide model selection in the presence of label noise. This approach can be used to select a high-performing model among different models, or to select a good checkpoint among different checkpoints of the same model throughout training. For example in Figure 3, we can observe that tracking susceptibility provides information about the epoch in the training process that the model is starting to memorize the noisy subset of the training set.
>
> --------------
> [1] LEARNING WITH NOISY LABELS REVISITED: A STUDY USING REAL-WORLD HUMAN ANNOTATIONS, Wei et al., ICLR 2022.
>
> [2] Part-dependent Label Noise: Towards Instance-dependent Label Noise, Xia et al., NeurIPS 2020.

---

> > ### Comment · Reviewer_QKAt · 2022-12-06
> > **Follow up**
> >
> > Thanks for the reply. My initial concerns about this paper are the threshold setting and the comparison to noisy validation set. The author's response has well addressed my concerns. Especially,  authors show that when the dataset size is small or the label noise level is high, the proposed susceptibility metric can outperform the use of a noisy validation set. I think it supports the efficacy of the proposed metric.
> >
> > I have increased my score to 6.

---

### Official Review · Reviewer_ZDzu · 2022-10-24

**Confidence:** 3
**Correctness:** 3
**Technical Novelty And Significance:** 2
**Empirical Novelty And Significance:** 2
**Recommendation:** 5

**Clarity, Quality, Novelty And Reproducibility:**

 - This paper has proposed a novel metric and the code has been attached. But the metric needs to be further verifed to prove its effectiveness in practice.


**Strength And Weaknesses:**

Strength:

 - Interesting and well-motivated idea. The proposed susceptibility is verified on many datasets including CIFAR, MNIST, and CLothing1m.Ting-ImageNet and Animal-10N.

 - This paper is well written and easy to follow

Weaknesses:

 - [1] already shows that the noisy validation is reliable. The authors state that [1] can not lead to comparing two trained models. However, from the experiments in [1], it seems that the models with higher acc on noisy validation also have higher acc on clean test acc. I think if the authors want to prove the efficiency of susceptibility. More experiments should be done to compare with [1]. From my perspective, it is hard to use susceptibility in practice since one also needs to set some threshold to remove models with higher susceptibility.


[1] Robustness of accuracy metric and its inspirations in learning with noisy labels. AAAI 2021.


**Summary Of The Paper:**

This paper proposes a new metric, called susceptibility, along with training acc to select good models trained on datasets with label noise. Authors observe that models with high training accuracy and low susceptibility will lead to higher acc on the clean test set. The authors also provide a convergence analysis of their methods.



**Summary Of The Review:**

 -I like the motivation. But currently, this paper fails to convince me that their metric is useful in practice. (See weakness).

---

> ### Author Response · Authors · 2022-11-15
> **Response to Reviewer ZDzu**
>
> We thank the reviewer for their feedback. Below we address the concerns you’ve raised, and we’d be very open to further discussion.
>
> **On comparison to a noisy validation set:**
>
> In our experiments in Table 1 and Figure 10, we observe that a high correlation between the noisy validation accuracy and the clean test accuracy is not always guaranteed for neither any amount of label noise level in the dataset or any size of the noisy validation set. In contrast, our approach consistently produces high values of correlations in all settings. In addition to robustness against label noise level and dataset size, it also offers a gauge on memorization, at a low computational cost without using additional data for this purpose. Please refer to the common response to all reviewers for more details on this comparison.
>
> **On finding thresholds in practice:**
>
> The thresholds are not manually set, e.g., by using hyper-parameter tuning, but are simply computed on the fly as the average values of the training accuracy and susceptibility over the available trained model. Therefore, setting these thresholds does not require any additional tuning, and does not bring any additional computational overhead.
> Please also refer to the common response to all reviewers for more details on these thresholds.

---

### Official Review · Reviewer_QseH · 2022-10-25

**Confidence:** 4
**Correctness:** 3
**Technical Novelty And Significance:** 3
**Empirical Novelty And Significance:** 3
**Recommendation:** 8

**Clarity, Quality, Novelty And Reproducibility:**

The paper is well-written, and the idea is clearly presented.

The proposed susceptibility metric is novel and of practical importance.

**Strength And Weaknesses:**

Strength:
- The proposed susceptibility metric is novel and interesting.
- Both empirical results and theoretical analysis are included to support the intuition that good models are resistant to memorization.
- The authors carried out extensive experiments to demonstrate the effectiveness of the proposed susceptibility metric.


Weaknesses:
- The discussion on different noise types is not sufficient. Only one kind of asymmetric noise is considered, and only the average test accuracy of the selected models is provided.  The basic idea of this paper is to connect the easiness of fitting one noisy signal (in the held-out set) and the memorization of the other noisy signal (noisy subset in the training set). Mostly, the paper assumes the two noisy signals are of the same type, i.e., produced by randomly labeling. However, it is likely the type of noisy signal could play a crucial role. If the noise type in the training set is completely different from the one in the held-out set, e.g., the feature-dependent noise, the corresponding susceptibility metric may not be able to reflect the memorization within the training set faithfully.
- The dependence of the effectiveness of the susceptibility metric on network architecture is not fully explored and discussed. For example,  the susceptibility metric is not that discriminative for the 5-layer CNN as shown in Fig.7 of the paper. Why is it so? In addition, it would be better to include more recently popular architectures such as the transformer.
- In section 4, the division of the train ACC - Susceptibility space into four regions is interesting. However, I did not find anywhere in the paper mentioning how to choose the threshold for the division.
- The discussion on different label noise levels is insufficient. The low label noise level situation deserves more emphasis since the noise levels of real-world datasets are not likely to be as high as 50%.
- Presentation:
    - Fig 1 needs better resolution.
    - Symbol [m] is used without a definition.
    - At the end of section 2, please indicate that the fully extended analysis is placed in section 5.
    - Symbol in Algorithm 1 could be confusing. t indicates the index of the epoch of normal training. However, at epoch $t$ the weight \tilde{W} is indexed with $t+1$. I understand that this refers to the “single optimization step” on $\tilde{S}$. Nonetheless, it is weird to have an index $t+1$ in the $t$-th epoch.


**Summary Of The Paper:**

This paper proposes a novel susceptibility metric to trace the memorization of noisy signals when the model is trained on a dataset with noisy labels. The computation of this metric does not require ground-truth labels. Both empirical evidence and theoretical analysis are presented to support the intuition behind this metric. Moreover, the authors conduct extensive experiments to demonstrate the effectiveness of this metric across different network architectures, datasets and noise types.

**Summary Of The Review:**

The finding of the relation between memorization and the easiness of fitting to a randomly-labeled held-out set is interesting from both theoretical and empirical perspectives. The proposed susceptibility metric is simple yet practical. Though the investigations could be extended in more depth, I would like to recommend the publication of this work.


------------------20221123----------------
As my concerns are all addressed by the helpful discussion with the authors, I decided to increase my rating from 6 to 8.

---

> ### Author Response · Authors · 2022-11-15
> **Response to Reviewer QseH (part 1)**
>
> We would like to thank the reviewer for their detailed review. We are glad that you find the paper novel and well-written and the solution of practical importance. Below we address the comments and questions you’ve raised, and we’d be very open to further discussion. We would appreciate it if you could increase your score if you are satisfied with our responses.
>
> **On studying different asymmetric label noise types:**
>
> In our submission, on top of one type of synthetic asymmetric label noise, we study datasets with real-world asymmetric label noises (Clothing-1M and Animal-10N) in Appendix E.1. The label noises in these datasets are not symmetric and are instance-dependent [2]. In addition, during the rebuttal phase, we have also included experiments on another real-world noisy labeled dataset CIFAR-10N [1], whose results are again consistently favorable (see Figures 27-29 (c)). The authors of [1] show both qualitatively and quantitatively that the label noise in this dataset is instance and feature-dependent. These experiments, carried out on 3 datasets with real-world asymmetric label noise, show that our approach works even for instance dependent and feature-dependent label noise types.
>
> **What if the noise type is different in the training set than the noise type to compute susceptibility?**
>
> This is indeed a very important question, which we investigate in our experiments with real-world noisy labeled datasets in Appendix E.1. In these experiments, the label noise in the datasets is not introduced by random labeling. Although susceptibility is indeed computed using random labeling, and thus without any information about the type of the label noise in the training sets, we can observe that it still tracks well memorization on these datasets. Our proposed model-selection method performs very well in this setting (Figure 29), consistently to the settings where the noise type is the same in the dataset and in the computation of susceptibility. We clarified this in the text of Appendix E.1 of the revised paper.
>
> **On different architectures:**
>
> First, we would like to point out that we have included recent popular architectures such as EfficientNet, RegNet, and MobileNet, even though not precisely transformers. Moreover, we can observe in Figure 3 that susceptibility is able to predict the memorization of different architectures. This is not limited to a certain architecture family: we study network architectures from shallow to deep and from residuals to VGGs. In addition, we can observe in Figure 30, that susceptibility is discriminative even among neural network architectures that differ only in their width and offer all a good resistance to memorization: Susceptibility to noisy labels $\zeta$ detects the most resistant model.
>
> **Susceptibility in CNNs:**
>
> Even for the simple CNN architecture, susceptibility is able to correctly track memorization in different models that differ in their regularization. We can observe in Figure 7 (top right) that a CNN without any regularization memorizes the noisy subset of the training set at the end of the training, whereas models that add regularization (both mixup and NCE) fix this issue. This is also precisely reflected in the value of susceptibility as well (bottom right) and there is a high correlation between memorization and susceptibility ($\rho=0.636$). Therefore susceptibility tracks memorization even across settings with the same architecture but which use different regularizations. We would be happy to answer any additional questions in this regard.
>
> **On how to set the thresholds:**
>
> The thresholds of these four regions are simply given by the average values of the training accuracy and susceptibility over the available models (this is mentioned in paragraph 3 of Section 4, and the caption of Figure 5 in the main paper). Please also refer to the common response to all reviewers for more discussion regarding these thresholds.
>
> **Low label noise levels:**
>
> In addition to studying low label noise levels in datasets with synthetic label noise, we study low label noise levels in real-world noisy labeled datasets as well. Our experiments on real-world label noise datasets have both a dataset with a low label noise level (Animal-10N has a label noise level of 8%) and a dataset with a higher label noise level and more complex data (Clothing-1M has a label noise level of 38.46%). Interestingly, we observe that our approach performs well for both datasets in Appendix E.1 and thus that susceptibility tracks memorization for both low and high label noise levels. Moreover, we have performed additional experiments during the rebuttal phase on another real-world noisy labeled dataset with a low label noise level (9%) called CIFAR-10N.  We observe consistently favorable results on this dataset as well, which we have included in Appendix E.1 of the revised paper (see Figures 27-29 (c)), confirming that our approach works also for low label noise levels.

---

> > ### Author Response · Authors · 2022-11-15
> > **Response to Reviewer QseH (part 2)**
> >
> > **Definition of symbol $m$:**
> >
> > Symbol $m$ is the number of hidden units in the neural network and is defined in paragraph 4 of Section 2.
> >
> > ------------------
> > [1] LEARNING WITH NOISY LABELS REVISITED: A STUDY USING REAL-WORLD HUMAN ANNOTATIONS, Wei et al., ICLR 2022.
> >
> > [2] Part-dependent Label Noise: Towards Instance-dependent Label Noise, Xia et al., NeurIPS 2020.

---

> > > ### Comment · Reviewer_QseH · 2022-11-21
> > > **about symbol [m]**
> > >
> > > my previous concern is about $[m]$ rather than $m$ itself. I assume $[m]:=\\{0,1,2,\dots,m-1\\}$.

---

> > > > ### Author Response · Authors · 2022-11-22
> > > > **Symbol [m]**
> > > >
> > > > Yes indeed, that's correct.

---

> > ### Comment · Reviewer_QseH · 2022-11-21
> > **Thanks for the reply; more discussions**
> >
> > I appreciate the detailed reply and am happy to see that most of my concerns are well addressed. I want to discuss further the  following two points:
> >
> > **1. when the noise in the training set differs from the one to compute susceptibility**
> >
> > I agree that experiments on real-world datasets support the claim that susceptibility can track memorization even when the noise types are different. However, the lack of precise knowledge about the noise in those real-world datasets weakens the argument. It would be desirable if one has controlled experiments on synthetic datasets with known noise of different types and intensities. In addition, the current results suggest that the tendency/ability to memorize, say, instance-dependent noise could be essentially the same as memorizing random noise. What do you think about this? Or Could the susceptibility show different behavior when the noise in the training sets introduces distinct patterns?
> >
> > **2. Susceptibility in CNNs**
> >
> > My previous concern is that susceptibility in CNN does not seem as discriminative as for other ARCH, e.g., resnext in fig.3. Further, I noticed the position of the turning point of the susceptibility curve is not that clear in fig. 7. Could this be due to the inductive bias of the underlying ARCH?

---

> > > ### Author Response · Authors · 2022-11-22
> > > **Reply to further discussions**
> > >
> > > Thanks for your reply and for bringing up these interesting discussions, which we'll respond to below.
> > >
> > > **1.** In our experiments with synthetic asymmetric label noise, we conducted two types of computations for susceptibility: (i) using random (symmetric) labeling, and (ii) using the same synthetic asymmetric label noise as in the training set. We observed in fact that the correlation between memorization and susceptibility is even higher in case (ii) than in case (i) (the Spearman correlation between memorization and susceptibility in case (i) is 0.57, and raises to 0.76 in case (ii)). But in all fairness, we did not want to include the stronger results of case (ii) in the paper,  to remain in the practical scenario of reference, where there is no access to the label noise type of the training set. In addition, there is a high correlation between susceptibility in case (i) and susceptibility in case (ii) (Spearman correlation is 0.89). This is why we only report our results based on computing susceptibility in case (i). We can add however the controlled experimental results for case (ii) in the final version of the work.
> > >
> > > As for the tendency of memorizing instant-dependent label noise and symmetric label noise, our experiments suggest that it is enough to use symmetric label noise to compute susceptibility (case (i) above) and track memorization accurately. They also suggest that a network that has not memorized instant-dependent noisy labels is also more resistant to memorization of random noise, than a network that has memorized instant-dependent label noise. This observation is in alignment with the theoretical and empirical results of Section 2 as well, namely that "good models are resistant to memorization". But indeed, as evidenced by the results mentioned above, a model that has memorized instant-dependent label noise would show less resistance to memorizing the same instant-dependent label noise, as opposed to memorizing some random label noise. As a result, the higher correlation between memorization within the training set and memorization of the held-out set is brought by using the same label noise type in the held-out set.
> > >
> > > **2.** Yes indeed, in Figure 3 some architectures with a sharp increase in their memorization  (for example resnext) show a clear increase in the value of their susceptibility as well. This parallel increase in memorization and susceptibility might be relatively less clear in other architectures (for example CNN) and the difference might indeed be due to the inductive bias: notice that resnext learns the clean subset of the training set also faster than CNN. Furthermore, for some architectures (for example shufflenetv2), we observe almost no increase in the values of memorization (train acc noisy) nor of susceptibility. We study architectures that have a relatively low memorization in more detail in Figure 30, and we still observe the high correlation between susceptibility and memorization.
> > >
> > > We would be happy to engage in further discussions that may come up.

---

> > > > ### Comment · Reviewer_QseH · 2022-11-23
> > > > **Thanks**
> > > >
> > > > Thanks for all those discussions and for bringing new results. I find them helpful and am satisfied that my concerns are addressed.
> > > >
> > > > The results convinced me that susceptibility is an effective and useful metric for model selection and could reveal some interesting behavior of deep learning models. I will increase my rating and look forward to seeing further future discussions on the mechanism of learning noisy signals by deep learning models.

---

### Official Review · Reviewer_E6P4 · 2022-10-25

**Confidence:** 3
**Correctness:** 3
**Technical Novelty And Significance:** 4
**Empirical Novelty And Significance:** 3
**Recommendation:** 6

**Clarity, Quality, Novelty And Reproducibility:**

The presentation is clear, but massive contents are left in the Appendix. It requires multiple jump back and forward for me to collect details and it might still be possible that some are missed. The originality and novelty is good.

**Strength And Weaknesses:**

Strengths:

- The paper is well-motivated and the presentation is clear.
- The experiments are extensive.
- Theoretical understanding is also provided.

Weakness & Questions:
- The method for picking a "good" model is slightly ad-hoc. The thresholds to separate regions are obtained by fitting a lot of models and then find the median. Therefore, for different datasets and models we need to repeat this process. More specifically, in Figure 5 the accuracy threshold is 40% and in Figure 15 it is 50%. The difficulties in deciding these values make the practical benefit of the proposed method questionable.
- What is the optimal value of "susceptibility" if we want to achieve good performance? It seems that there would be an optimal range, but is it possible to clearly find such a point that can achieve the best performance?  Ultimately, can the "susceptibility" be a predictive indicator for the performance?
- Can we use "susceptibility" to regularize the models' training rather than just observe?

**Summary Of The Paper:**

This paper proposes a metrics called "susceptibility" that measure the model's resistance to memorization by using randomly labeled data. The authors provide both theoretical and empirical observations that the models which are resistant to memorization will have high test accuracies. Therefore, the extent of memorization to noisy labels can be used for picking models. The authors demonstrate how to pick models based on the values of susceptibility and training ACCs.


**Summary Of The Review:**

This paper introduces an interesting metric, and the authors justify that the metric is useful for selecting models. However, I feel the correlation or the relationship between the metric and the final performance is vague (or at least need to be better illustrated).

---

> ### Author Response · Authors · 2022-11-15
> **Response to Reviewer E6P4**
>
> We thank the reviewer for their review. We are glad you find the importance, novelty, and clarity of our work significant. Below we address the comments and questions you’ve raised, and we’d be very open to further discussion.
>
>
> **On the different thresholds:**
>
> Because of the difference in the complexity of datasets in Figures 5 and 15 (MNIST is less complex compared to CIFAR-10), desirable models in each dataset have different levels of *trainability* and *memorization* and hence different values of training accuracy and susceptibility. Because of the lack of information regarding the level of trainability and memorization, these thresholds cannot be set a priori for a new dataset and a new set of models. Yet, we emphasize that setting these thresholds for a new setting does not introduce any computational overhead to the model selection process. Our proposed model selection approach works even with only one single model being trained (by considering different checkpoints of the model instead of different models). As in any model selection approach, the larger the number of choices for models are, the better the selected model is.
> Please also refer to the common response to all reviewers for more discussion on the thresholds.
>
>
> **Can susceptibility predict performance? Is there a correlation between susceptibility and performance?**
>
> The susceptibility metric is strongly positively correlated with memorization on the training set (see for example in Figures 3, 7, and 30-33). Memorization on the training set degrades the predictive performance of the model on unseen clean data. Hence, given multiple models with the same overall training accuracy, susceptibility is predictive of the amount of memorization and therefore of performance.
>
>
> **On the optimal value of susceptibility:**
>
> Susceptibility is a metric that depends on the given dataset. This dependence is indeed expected, because susceptibility is a dataset-specific feature: it tracks the memorization of models on a given dataset, and memorization might change with a different dataset and model. As a result, we don't have a globally optimal value for this metric. We propose susceptibility as a metric that relatively tracks memorization for different checkpoints of the same model, or for different models.
>
>
> **On using susceptibility as a regularization:**
>
> This is indeed a very interesting suggestion and direction to investigate for future work. In particular, it would be interesting to explore how one can backpropagate through a measure of susceptibility to minimize memorization.
>
>
> **On the contents being left in the appendix:**
>
> We agree that some details had to be left in the appendix because of the page limit. We hope to have placed the most important content in the main body. We would highly appreciate any explicit suggestions in this regard.

---

### Author Response · Authors · 2022-11-15
**Common response to all reviewers**

We are very thankful to all reviewers for their time. We are glad that they appreciated the motivation for introducing susceptibility as an easy-to-compute metric, the theoretical foundations, and the extensive experiments supporting it. Below we address some common concerns shared among more than one reviewer. We would be happy to respond to any additional questions that may come up.

**On finding thresholds in practice:**

Setting the thresholds to find different regions is practically straightforward because it does not require any prior knowledge on the dataset, nor any additional computations or tuning of any kind. We would like to emphasize that tuning these thresholds could result in even higher test accuracies for the selected models (as pointed out by the study in Appendix F and Figure 39 of the revised paper). However, we want to remain in the practical setting where we do not have any access to a clean validation set for tuning. Therefore, we avoid any hyper-parameter tuning. And indeed, throughout our experiments, these thresholds are never tuned nor set manually to any extent. Among thresholds that can be computed without access to a clean validation set, we opted for the average values of susceptibility and training accuracy (over the available models) for simplicity and efficiency (more details can be found in Appendix F and Figure 39 of the revised paper).

**On comparison to a noisy validation set:**

Below we list the advantages of our approach for selecting models using a noisy validation set (a comprehensive study is also carried out in Appendix C):

1. A gauge on memorization:
Tracking memorization is more difficult than simply tracking overfitting, as there might be different reasons behind the overfitting, memorization being only one of them. Our approach provides additional information on memorization within the training set, which is even absent from the information brought by a test set or a noisy validation set.
For example, Figure 35 (a) shows a model on the very top right with a high test accuracy and 100% memorization of the noisy subset. However, some models (on the top middle/left part of the figure), which have the same fit on the clean subset but lower memorization, have lower or similar test accuracy. Hence, even the clean test accuracy was not able to compare these models in terms of memorization let alone the accuracy on a noisy validation set.

2. Robustness against label noise level and dataset size:
The experiments reported in Appendix C demonstrate that susceptibility clearly outperforms the use of a noisy validation set when the dataset size is small and/or its label noise level is high (Table 1 and Figure 10). When the label noise level is low and the dataset is large, the performance of both approaches is comparable (Figure 10).
As formally discussed in [1], the number of noisy validation samples that are equivalent to a single clean validation sample depends on the label noise level of the dataset and on the true error rate of the model, which are both unknown in practice. In contrast, our approach does not need to know them and is robust against label noise level and dataset size.

3. No need for extra data:
As pointed out in item 2 above, depending on the label noise level of the given dataset, the amount of samples required to be in the noisy validation set in order to have a high and positive correction might be large. Hence, a possibly large subset of the available dataset needs to be set aside: Increasing the size of the validation set removes data out of the training set, which degrades the overall performance. In contrast, our approach leaves the entire available dataset for training.

4. Low computational cost:
In Table 1, we can observe that to reach the same correlation value, a validation set needs to be around ten-fold of the set used to compute susceptibility. This shows that susceptibility requires much less computation as well (throughout the paper we only take a single mini-batch to compute it).

-----------------
[1] Evaluating Classifiers by Means of Test Data with Noisy Labels, Lam and Stork, IJCAI 2003.

---

### Decision · Program_Chairs · 2023-01-20

**Decision:**

Accept: poster

**Justification For Why Not Higher Score:**

The submission is not more competitive since its problem of interest (i.e., memorization under non-class-conditional label-noise learning) is very specific.

**Justification For Why Not Lower Score:**

Although the idea is simple, it is novel and works in practice. The empirical side is quite strong.

**Metareview: Summary, Strengths And Weaknesses:**

The paper studied how to characterize and track memorization of data in label-noise learning. It proposed an effective metric called "susceptibility" which relies on unlabeled data and measures memorization of randomly labeled data. The proposal can be applied in validation/hyperparameter-tuning as a replacement of noisy validation data identically distributed with noisy training data. Although the idea is simple, it is novel and works in practice. Therefore, we should accept it for publication. Please carefully prepare the camera-ready version by taking the comments from the reviewers (especially on controlled experiments).

**Note From Pc:**

if the above contains the word "oral" or "spotlight" please see: "oral" presentation means -> notable-top-5% and "spotlight" means -> notable-top-25%. As stated in our emails, we are disassociating presentation type from AC recommendations

**Summary Of Ac-Reviewer Meeting:**

Two reviewers, QseH and QKAt, attended the meeting. We exchanged the opinions fruitfully. The theoretical side is not strong but the empirical side is quite strong. We all vote for acceptance. BTW, our reviewers complained that the authors insisted their method was evaluated on real-world datasets and no need to do so on benchmark datasets under controlled instance-dependent noise, and then I pushed them to do so in my meta-review to make their experiments more comprehensive.

Another positive reviewer, E6P4, sent an email to me and also supports the paper: "I am really sorry but I got COVID affected and not able to speak without coughing at this moment. I waited until tonight but the situation did not improve. Therefore I am writing to skip this meeting, but I would send my updated response tomorrow (I will still support this paper)."